

# Auditory processing in the zebra finch midbrain: single unit responses and effect of rearing experience

Priscilla Logerot[1], Paul F. Smith[2], Martin Wild[3] and M. Fabiana Kubke[4]

[1] Anatomy and Medical Imaging, University of Auckland, University of Auckland, Auckland, New Zealand

[2] Dept. of Pharmacology and Toxicology, School of Biomedical Sciences, Brain Health Research Centre, Brain Research New Zealand, and Eisdell Moore Centre, University of Otago, Dunedin, New Zealand

[3] Anatomy and Medical Imaging and Eisdell Moore Centre, University of Auckland, University of Auckland, Auckland, New Zealand

[4] Anatomy and Medical Imaging, Centre for Brain Research and Eisdell Moore Centre, University of Auckland, University of Auckland, Auckland, New Zealand

Corresponding author
M. Fabiana Kubke,
f.kubke@auckland.ac.nz

## ABSTRACT

In birds the auditory system plays a key role in providing the sensory input used to discriminate between conspecific and heterospecific vocal signals. In those species that are known to learn their vocalizations, for example, songbirds, it is generally considered that this ability arises and is manifest in the forebrain, although there is no a priori reason why brainstem components of the auditory system could not also play an important part. To test this assumption, we used groups of normal reared and cross-fostered zebra finches that had previously been shown in behavioural experiments to reduce their preference for conspecific songs subsequent to cross fostering experience with Bengalese finches, a related species with a distinctly different song. The question we asked, therefore, is whether this experiential change also changes the bias in favour of conspecific song displayed by auditory midbrain units of normally raised zebra finches. By recording the responses of single units in MLd to a variety of zebra finch and Bengalese finch songs in both normally reared and cross-fostered zebra finches, we provide a positive answer to this question. That is, the difference in response to conspecific and heterospecific songs seen in normal reared zebra finches is reduced following cross-fostering. In birds the virtual absence of mammalian-like cortical projections upon auditory brainstem nuclei argues against the interpretation that MLd units change, as observed in the present experiments, as a result of top-down influences on sensory processing. Instead, it appears that MLd units can be influenced significantly by sensory inputs arising directly from a change in auditory experience during development.

## INTRODUCTION

In songbirds, as in many other vertebrates, vocal signals play a key role in communication (*Catchpole & Slater, 2003*). Most vertebrates use vocal communication signals that are innate and supported by a suite of interconnected brainstem nuclei that contribute to the

production and identification of conspecific calls (*Kennedy, 1974*; *Kubke & Wild, 2018*). A handful of lineages (including songbirds) have, in addition to the innate repertoire, vocalizations that are learned during a critical developmental period (*Zann, 1990*; *Bolhuis & Moorman, 2015*). In songbirds these vocalizations are learned first by memorising the song of a tutor (usually the father) during early life, and by later learning to match their vocal output to the internal template of the song (*Bolhuis & Moorman, 2015*). A set of forebrain nuclei collectively known as the 'song system' are, under the influence of auditory input, involved in the learning, production, and maintenance of the song (Fig. 1A) (*Brainard & Doupe, 2000*). Most studies on auditory processing in songbirds have focused on forebrain areas; the role of the auditory brainstem in extracting auditory information used in the learning, production and maintenance of song is less well known.

A key role of the auditory system is to mediate the identification of conspecific vocal signals (both learned and innate) from other environmental sounds (including heterospecific vocalizations). A substantial body of work in other vertebrates has shown selectivity for innate communication sounds in the auditory midbrain where neurones show finely tuned responses to spectral and temporal features of species-specific vocal communication signals (*Scheich, Langner & Koch, 1977*; *Feng, Hall & Gooler, 1990*; *Bodnar et al., 2001*; *Bass & McKibben, 2003*; *Šuta et al., 2003*; *Bass, Rose & Pritz, 2005*; *Covey & Carr, 2005*; *Portfors & Sinex, 2005*; *Rose & Gooler, 2007*; *Rodríguez, Read & Escabí, 2009*; *Holmstrom et al., 2010*; *Wilczynski & Ryan, 2010*; *Rose, Leary & Edwards, 2011*; *Sayegh, Aubie & Faure, 2011*; *Pollak, 2013*). In songbirds, numerous studies describe auditory processing of the learned song primarily in the auditory forebrain, where different regions are suggested to encode different aspects of the perception and selection of conspecific signals (for review see *Knudsen & Gentner, 2010*). These nuclei receive the necessary auditory input indirectly by way of ascending inputs from the brainstem auditory nuclei and thalamus (Fig. 1B; *Nottebohm, Stokes & Leonard, 1976*; *Nottebohm, Paton & Kelley, 1982*). Studies on the role of the lower auditory system in general, and the midbrain in particular, in the processing of innate and learned vocal signals are lacking or limited, despite the fact that the auditory midbrain of songbirds, like that of other vertebrates, is well positioned to serve as a major centre where selectivity to learned vocal signals could arise (*Feng, Hall & Gooler, 1990*; *Bass, Rose & Pritz, 2005*; *Covey & Carr, 2005*; *Portfors & Sinex, 2005*; *Rose & Gooler, 2007*; *Wilczynski & Ryan, 2010*; *Vonderschen & Chacron, 2011*; *Wenstrup, Nataraj & Sanchez, 2012*).

The major contribution to our understanding of auditory processing in the songbird auditory brainstem has come from the work of Woolley and colleagues in the zebra finch, who showed that neurones in the auditory midbrain (mesencephalicus lateralis pars dorsalis, MLd) are tuned to specific spectro-temporal modulations of the conspecific song, suggesting that in birds, too, the auditory midbrain plays a central role in the processing of communication signals (*Woolley & Casseday, 2004*; *Woolley et al., 2005*, *2009*; *Woolley, Gill & Theunissen, 2006*; *Woolley, Hauber & Theunissen, 2010*; *Schneider & Woolley, 2010*; *Woolley, 2012*). Further, the work of *Woolley, Hauber & Theunissen (2010)* also showed that the responses found in the auditory midbrain could be modified by early

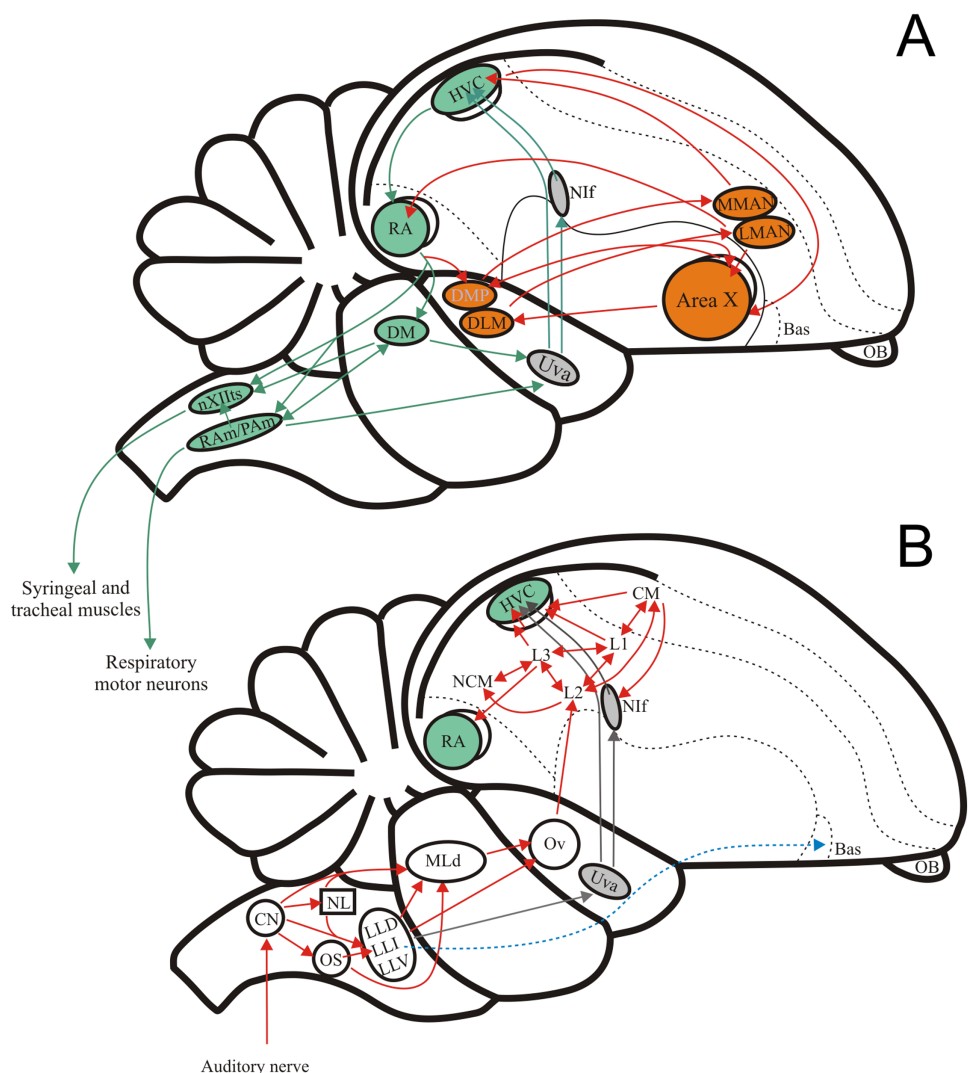

**Figure 1 The avian song system and the avian ascending auditory system.** (A) Schematic showing the premotor pathway involved in vocal output (nuclei and connexions labelled in green) and the Anterior Forebrain Pathway involved in song learning (nuclei and connexions labelled in orange). (B) Schematic showing the ascending connexions of the auditory pathway. Grey arrows indicate a parallel pathway for auditory input into the song system. Abbreviations: HVC, proper name; RA, robustus archopallialis; DM, dorsal medial nucleus of the intercollicular complex; nXIIts, trachosyringeal portion of the hypoglossal nucleus; RAm/Pam, nuclei retroambigualis/parambigualis; Uva, n. uvaeformis; NIf, n. interface; MMAN, medial portion of the magnocellular n. of the anterior nidopallium; LMAN, lateral portion of the magnocellular n. of the anterior nidopallium; AreaX, proper name; DMP, dorsomedial nucleus of the posterior thalamus; DLM, medial portion of the dorsolateral anterior thalamic nucleus; Bas, basorostral n. of the pallium; OB, olfactory bulb; CM, caudo- medial mesopallium; NCM, caudal medial nidopallium; L1, proper name; L2, proper name; L3, proper name; Ov, ovoid n. of the thalamus; MLd, n. Mesence-phalicus lateralis pars dorsalis, i.e., auditory torus; LLD, dorsal n. of the lateral lemniscus; LLI, inter-mediate n. of the lateral lemniscus; LLV, ventral n. of the lateral lemniscus; OS, superior olive; NL, n. laminaris; CN, cochlear nuclei, i.e., n. magnocellularis and n. angularis.

developmental experience, thus linking the midbrain to the filtering features that support song learning.

*Woolley, Hauber & Theunissen (2010)* exploited the changes in song structure and behavioural preference that follow the cross fostering of zebra finches by Bengalese finches (*Campbell & Hauber, 2009*). When zebra finch chicks are cross-fostered by Bengalese finches they grow to produce a song that: (1) incorporates elements of the foster parents; (2) contains more elements than the normal zebra finch song but fewer than the Bengalese finch song; and (3) is longer than the normal zebra finch song but shorter than that of the foster male (*Eales, 1987*; *Clayton, 1989*; *Takahasi et al., 2006*).

*Woolley, Hauber & Theunissen (2010)* compared the responses to zebra finch and Bengalese finch songs in the MLds of normal reared zebra finches, and zebra finches that had been cross-fostered by Bengalese finches. They report that presentation of zebra finch or Bengalese finch songs in normal-reared and cross-fostered birds appear to evoke similar firing rates in MLd units, even though, overall, the mean firing rate is diminished in cross-fostered birds. Mutual information rates were also similar when comparing responses to zebra finch or Bengalese finch songs in both normal-reared and cross-fostered birds. This is in contrast to behavioural studies (*Campbell & Hauber, 2009*) in which normal reared zebra finches were shown to 'prefer' zebra finch song over Bengalese finch song, whereas cross-fostered zebra finches associated equally with both types of songs. Taken together, these results would suggest that the different behavioural preferences to zebra finch or Bengalese finch song in normal reared and cross fostered birds cannot be accounted for by response differences in MLd.

Using a similar paradigm to that of *Woolley, Hauber & Theunissen (2010)*, we here examine the responses of units in MLds to zebra finch and Bengalese finch songs in normal-reared and cross fostered zebra finches (*Campbell & Hauber, 2009*, *2010*). In contrast to Woolley et al.'s findings, we find that units in MLd change their response properties based on developmental experience. Our data indicate that units in MLd of normally-reared zebra finches show a bias toward conspecific (zebra finch) song, whereas in zebra finches cross-tutored by Bengalese finches, the bias of MLd units towards zebra finch vocalizations was reduced. Thus, our findings suggest that the ontogenetic changes that affect behavioural preference to conspecific over heterospecific song are paralleled by neuronal responses recorded from MLd.

## METHODS

### Animals and ethical considerations

The experimental work was carried out in compliance with and approval from the University of Auckland Animal Ethics Committee (R425) in accordance with the University of Auckland Code of Ethical Conduct for the Use of Animals for Teaching and Research, the Animal Welfare Act 1999 (New Zealand), and The National Animal Ethics Advisory Committee (NAEAC) Good Practice Guide for the Use of Animals in Research, Testing and Teaching. Birds were provided with ad-lib water and commercial dry finch-mix seeds (Animates™), spray millet and additional weekly treats (fresh grasses, corn boiled egg, and rice). Birds were also supplied with calcium supplements and grit.

## Normal reared zebra finches

Six adult zebra finches (2 males and 4 females >100 days old) that were raised having been exposed only to their conspecifics were used in this study. Birds were bred in an aviary at the Department of Psychology, University of Auckland (*Campbell & Hauber, 2010*). Light was provided via a compact Arcadia fluorescent bird lamp (2.4% UVB and 12% UVA) which imitated outdoor conditions. The mean temperature in the aviary was kept at 21 ± 3 °C, daily humidity between 35% and 65% and care was taken to keep a constant airflow. Pairs of zebra finches (*Taeniopygia guttata*) were housed together in indoor cages and once the chicks reached adulthood (>100 days), they were transferred to a holding indoor aviary (2 m × 1 m × 2 m) in the Faculty of Medical and Health Sciences. At all times, birds were held at a constant light/dark cycle of 14 h/10 h in the aviary and birds were provided with food and water ad lib. Thus, the ZF-ZF group had been only exposed to conspecific song.

## Cross fostering

Six male zebra finches (>100 days old) that were raised by Bengalese finches (ZF-CF) were used. These zebra finches had previously been used in behavioural experiments and a full description of the rearing conditions is available in *Campbell & Hauber (2009)*. Briefly, zebra finch chicks (3–5 days old) were transferred into the nest of a Bengalese finch pair in one of two outdoor aviaries in Auckland (2 m × 1 m × 2 m), and exposed to natural photoperiod and weather conditions. The chicks were therefore reared hearing male and female Bengalese finches and tutored by a Bengalese finch male, but were also exposed to wild birds species found commonly in suburban Auckland (Blackbird *Turdus merula*; House Sparrow *Passer domesticus*; Song Thrush *Turdus philomelos*; Starling *Sturnus vulgaris*; and Tui *Prosthermadera novaeseelandiae*). After reaching adulthood (>100 days), the cross-fostered birds were transferred into single-sex aviaries where they were housed with fostered conspecifics. After the behavioural experiments were completed, the birds were finally transferred to an indoor aviary (2 m × 1 m × 2 m) at the Faculty of Medical and Health Sciences, co-housed with the group of normal reared zebra finches. At all times, birds were held at a constant light/dark cycle of 14 h/10 h in the aviary and birds were provided with food and water ad lib. These birds did not hear normal conspecific vocalisations before behavioural testing, at which time their songs were crystallised.

## Surgery

Birds were deeply anesthetised with an intramuscular injection of a mixture consisting of 55 mg/kg of ketamine (Parnell Laboratories, Auckland, New Zealand) and 11 mg/kg Xylazine (Rompun, Bayer) delivered in a volume of 0.04–0.05 ml. Additional small doses of anaesthetic were given as required during the course of the experiment. The head was fixed in a custom-made stereotaxic apparatus (Herb Adams, Los Angeles, CA, USA) with ear and beak bars and the head was tilted down 28° from the vertical axis. The head skin was reflected from the midline, the skull surface cleaned and a small metal plate was fixed to the skull using dental cement (Land Dental Manufacturing Co., Inc., Wheeling, IL, USA). This allowed the head to be held at a fixed angle during experimentation after

removing the ear bars to present auditory stimulation. The coordinates for electrode insertion were measured with respect to the bifurcation of the mid-sagittal sinus (Y sinus). A small opening in the skull and dura mater was made over the chosen coordinates to allow electrode penetration into the midbrain.

## Stimulus generation and presentation

Auditory stimuli consisted of pure tones, white noise, conspecific and heterospecific songs. Both white noise and tones were generated using Adobe Audition 3.0 software taking into account described hearing ranges for zebra finches (see *Okanoya & Dooling, 1987*). Pure tone stimuli (1, 2, 3, 4, 5 and 6 kHz ) were of 1 s duration with built-in rise and fall ramps of 150 ms at onset and offset to avoid stimulus clicking through the speaker. The white noise (WN) stimulus was also 1 s in duration and contained frequencies ranging from 0 to 10 kHz. Noise and tone stimuli were presented at 75 dB SPL, as measured with an SPL metre (RadioShack) 30 cm from a free-field speaker.

The songs used were conspecific (zebra finch) and heterospecific (Bengalese finch) songs. The conspecific songs were chosen from birds raised in the same aviary as the birds used in this experiment and were therefore familiar to both ZF-ZF and ZF-CF birds. They were obtained by recording vocalizations of each individual bird using Sound Analysis Pro software. They were then bandpass filtered (300 Hz–12 kHz) using Raven 1.3 software. Heterospecific songs were kindly provided by Dr. Sarah Woolley from Columbia University, New York. Song lengths ranged between 1.816 s and 2.691 s for the conspecific songs and 1.940 s to 3.091 s for the Bengalese finch songs. All songs were calibrated to be presented with an average power of 75 dB (as calibrated using a ½-inch free-field microphone, Brüel and Kjaer and Raven 1.3 software). Songs were played in the forward and backward directions.

Each stimulus (white noise, tones and songs) was constructed so as to contain 500 ms of silence preceding and following the actual stimulus. Each stimulus was presented to the bird 15 consecutive times, with different stimuli presented in a randomised order. The interval between the 15 consecutive presentations of a single stimulus type was kept constant within a sequence, but these intervals were randomised for each separate sequence (between 0.5 and 1.5 s, 0.1 s steps). Each unit therefore received 19 stimuli: white noise, 6 pure tones and 12 songs (3 conspecific songs in their forward and reverse directions and 3 heterospecific songs in their forward and reverse directions). At the end of the presentation of each set of stimuli, 30 s of baseline activity for each unit was also recorded at least 2 min after the end of the last stimulus presentation, and this activity was used to calculate the spontaneous firing rate.

Stimuli were presented to the bird through a multifunction processor (TDT System 3; Tucker-Davis Technology, Alachua, FL, USA), routed through an attenuator (TDT) that allowed equal intensity presentation of all stimuli at 75 dB SPL average power. The signal was then processed through a TDT Stereo Amplifier connected to a free-field magnetic loudspeaker (TDT) placed 30 cm in front of the bird. Before an experiment, the output of the loudspeaker was checked using a RadioShack SPL metre so to ensure constancy of the average power at which songs were delivered.

## Electrophysiological recordings

All recordings were made in a sound attenuation chamber (Microbooth, All-DUCT Fabrication, PTY. Ltd, Melbourne, VIC, Australia), using tungsten or platinum backed electrodes (FHC Inc., Bowdoin, ME, USA; impedances ~10 MΩ at 1 kHz). The location of MLd was determined using a variety of search stimuli (white noise, hand claps, clicks, vocalizations), so as to avoid any bias that could result from missing cells responsive to a narrow range of stimuli. Tones were not used as search stimuli, so the data set may contain an underrepresentation of units responsive only to specific tones. The recorded signal was filtered (300 Hz low/5 kHz high pass) and amplified (gain 100×) using an AC amplifier (A-M systems Model 1800) and digitised using the TDT multifunction processor RX6. Threshold and spike discrimination were achieved in real time using the OpenController interface of the TDT OpenEx Suite. Raster plots and peristimulus-time histograms (PSTH) were visualised in real-time using the OpenScope interface, for a rapid qualitative assessment of the auditory selectivity of isolated units prior to the stimulus presentation protocol. The sequence of song presentations and unit recordings were made using a custom-made programme designed using the TDT RPvdsEx control (P. Logerot). During the experiment, individual stimuli were chosen from a prebuilt collection of stimuli through the OpenController interface, which also allowed us to set the number of consecutive presentations of the stimulus (15 for this study), the inter-stimulus interval and the attenuation. The order in which stimuli were presented was determined randomly for each series of presentations. The epoch store tracked the onset of each stimulus repeat within a stimulation sequence, and this was then used to align sequential presentations to construct the peristimulus-time histograms and raster plots.

Electrolytic lesions (40 μA, DC for 10 s) were made to identify and reconstruct recording site locations but, unfortunately, these proved frequently too large to assess the sites with precision. They could, however, be confidently used to confirm the recording site within MLd.

## Analysis

Units were analysed off-line using TDT's OpenSorter and OpenExplorer packages of the OpenEx suite. OpenSorter was used to manually eliminate outlier spikes and to confirm that recordings represented single units. When in doubt, the data were not included in the final analysis. The spontaneous firing rate of the unit (spikes/s) was calculated using the first 20 s of the 30 s baseline activity recorded at the end of the session.

Units were classified as 'auditory' when they reached criterion for at least one of the stimuli (whether WN, a tone or a song). Post-stimulus histograms (PSTHs) of 20 ms bins were constructed using TDT Openexplorer. Neurons were considered to reach the criterion if the responses within at least one 20 ms bin were above threshold, with threshold set at the mean +5 S.D. of the baseline firing rate (defined as the firing rate during the 500 ms of silence preceding the sound stimulus) (*Prather et al., 2009*). This choice of threshold biases the population used for analysis, since units that show an inhibition as a result of the auditory stimulus would not be included in the analysis, unless

they reached criterion when presented with another stimulus. Once a unit met the criterion for at least one stimulus, the responses to all of the stimuli were considered for the analysis.

### Response strength

Response strength (RS) measures the amount of evoked activity above spontaneous rate. RS is sensitive to changes in levels of evoked activity in response trains that are sustained during the presentation of the stimuli and is also sensitive to inhibitory response. RS was measured for all song stimuli (conspecific, heterospecific, forward and reverse) for all units that met the criteria to be classified as auditory (see above). Response strength was calculated for each song by subtracting the spontaneous activity (in spikes/s, spontaneous spike rate (SSR)) from the evoked spike rate (spikes/s, evoked spike rate (ESR)) for each of the 15 consecutive presentations and averaging these values:

$$RS = \frac{\sum_{i=1}^{N} ESRi - SSRi}{N}$$

where RS is the Response Strength (spikes/sec), ESR is the evoked spike rate, SSR is the spontaneous spike rate and $N$ is the sample size in terms of the number of stimuli. RS values have the disadvantage of not distinguishing changes in the timing of action potentials that may not be accompanied by overall changes in firing rate. Thus, low values of RS could indicate both a lack of response as well as a highly feature-specific response. For example, a unit could show small changes of RS around threshold, yet show a clear change in the temporal pattern of the spikes associated with the song stimulus.

### Selectivity index (SI) and d′

To compare unit responses between two stimuli we used the Selectivity Index (SI) and $d'$. Both SI and $d'$ values can be used to assess a unit's selectivity (or 'preference') of one stimulus over another in the pair. Both quantifiers are based on the unit's mean response strength (RS) values for each of the stimuli (and therefore will be influenced by 'inhibited' or time-locked responses). Unlike the comparisons based on the selectivity index, the $d'$ does not provide information regarding the strength of the responses themselves.

SI compares the responses of a neurone to two different stimuli ($A$ and $B$) but does not take into account response variability. The SI is calculated as:

$$SI = \frac{RS_A}{RS_A + RS_B}$$

$d'$ measures the difference in response (or preference of a unit) to a pair of stimuli, $A$ and $B$. If the value of $d'$ (of responses to $A$ compared to $B$) is greater than zero, it indicates that stimulus $A$ elicited a greater response in the unit than did stimulus $B$. A $d'$ smaller than zero, indicates that stimulus $B$ elicited a larger response than stimulus $A$, and values equal to zero indicate that the unit responded equally to both stimuli. Differences in the responses to two stimuli ($A$ and $B$) are calculated by comparing the RS of the unit to each of the two stimuli ($RS_A$ and $RS_B$, respectively).

The $d'$ is calculated as:

$$d'_{(A \to B)} = \frac{2(\text{mean RS}_A - \text{mean RS}_B)}{\sqrt{\sigma^2 \, \text{RS}_A + \sigma^2 \, \text{RS}_B}}$$

where $d'$ is the discriminability/selectivity in response to a pair of stimuli, $A$ and $B$, $\text{RS}_A$ and $\text{RS}_B$ is Response Strength to stimuli $A$ and $B$, respectively, and $\sigma^2$ is the variance. One advantage of the $d'$ value is that the differences between the means of the RS for each stimulus is weighted against the variance of their distributions. A second advantage is that the $d'$ is insensitive to sample size.

Which values of SI and $d'$ would be considered as the criterion for selectivity was determined in a way similar to that of *Doupe & Solis (1997)*. For SI, a unit is considered to be selective when the mean RS of the unit to one stimulus is at least twice that of the other stimulus (i.e., with SI = <0.33 or SI> = 0.66). For $d'$ a non-selectivity zone interval was chosen in which most neurones showed non-selectivity based on the SI criterion. A criterion of $|d'| = 2$ was thus set.

## Generalized linear mixed model

We used a Generalized Linear Mixed Model (GLMM) in SAS 9.3 and SPSS 26 for the analysis of the data. This analysis took into account the lack of independence of the individual data points from multiple neurones recorded from different animals, which otherwise may have led to pseudoreplication (for reviews on these issues, see *Lazic, 2010*; *Nakagawa & Hauber, 2011*). The analysis took into account the following: (1) That each isolated single unit in this experiment was presented with 19 different stimulations each repeated 15 times, that is the recording session presented 285 stimuli; and (2) that several units were recorded from a single animal.

The data were first tested for the normality of the residuals using the Shapiro–Wilk test and then natural log transformed in an attempt to rectify their non-normal distribution. Homogeneity of variance for the residuals was tested using the Levene test and it was confirmed that this assumption was satisfied. The Levene test was 0.13 before transformation and 0.42 after a natural log transformation; therefore, the null hypothesis of homogeneity of variance was retained.

In order to investigate factors influencing the response of units within birds, a GLMM was fitted with bird and unit within bird as random effects to allow for the clustering of units within birds. Bird category was a fixed, between group factor, and song ID, conspecific/heterospecific (CON/HET) and song direction (i.e. song played either forwards or backwards; FOR/REV) were designated as repeated measures using an unstructured covariance matrix model. The GLMM was conducted using a restricted maximal likelihood estimation (REML) procedure and the Kenwood-Rogers approximation for the degrees of freedom (*Gurka & Edwards, 2007*; *McCulloch, Searle & Neuhaus, 2008*). The best covariance matrix model (i.e. unstructured) was chosen on the basis of the smallest Akaike Information Criterion (*Gurka & Edwards, 2007*; *McCulloch, Searle & Neuhaus, 2008*; *Field, 2011*). The interactions of bird category, song type, CON/HET and song direction, were initially included as explanatory variables.

The general equation (in matrix notation) for the GLMM analysis used was:

$$y = X\beta + Zu + \varepsilon$$

in which $y$ = an $N \times 1$ column vector of outcome variables; $X$ = an $N \times p$ matrix of $p$ predictor variables ($X$ included Bird Category, Song type (CON/HET), direction (FOR/REV), Song ID and Bird Category $\times$ CON/HET); $\beta$ = a $p \times 1$ column vector of fixed effects regression coefficients; $Z$ = an $N \times q$ matrix of the $q$ random effects related to a fixed $\beta$ ($Z$ included factors Bird ID and unit within bird); $u$ is a $q \times 1$ vector of random effects; and $\varepsilon$ = an $N \times 1$ column vector of residuals (*Gurka & Edwards, 2007*; *McCulloch, Searle & Neuhaus, 2008*). Initially, the full factorial model was fitted but then non-significant interactions were systematically omitted in search of the best model for the data, as indicated by the lowest AIC (*Rouder et al., 2016*). This was a model using unstructured covariance, excluding all interactions except for Bird Category $\times$ CON/HET. As with regression, it is appropriate to remove non-significant terms in search of the best model for the data, providing that some independent criterion such as the AIC is used as a guide (*Gurka & Edwards, 2007*; *McCulloch, Searle & Neuhaus, 2008*; *Rouder et al., 2016*).

Post-hoc comparisons were conducted using the least square means estimates from the GLMM (*Field, 2011*). Mann–Whitney $U$ tests were used to compare the distributions and medians of neuronal responses (*Field, 2011*).

In normal reared zebra finches we found that for both sexes there was evidence of a greater response to CON than HET songs (males $P = 0.001$, females $P = 0.0003$) and that the size of this difference was greater in males than females. Therefore, sex was not introduced in the analysis, since it is unlikely that the uneven distribution of sex between the two groups would influence our results. This is consistent with behavioural data where sex was not found to be a factor in similar rearing experiments (*Campbell & Hauber, 2009*).

## RESULTS

We sought to describe the basic responses of single units in MLd to simple stimuli (tones and white noise) and to conspecific (zebra finch) and heterospecific (Bengalese finch) song in normal-reared zebra finches, and in zebra finches that were cross-fostered by Bengalese finches. We present the results for those MLd units that reached criterion to be classified as auditory when all stimuli (white noise, tones, and songs, both conspecific and heterospecific) were considered (58 units from normal reared bids, ZF-ZF: 6 birds, 4–14 units/bird; and 20 units in the cross-fostered birds, ZF-CF: 6 birds, 1–6 units/birds, with 1 ZF-ZF unit where responses to HET-3 is missing). In neither control nor cross-fostered groups was there an obvious individual bird effect on the units' response.

### Spontaneous activity and responses to tones and white noise

Most units in the MLd of ZF-ZF and ZF-CF showed no spontaneous activity (<1 spike/sec) or showed low spontaneous activity (<5 spikes/s) (66% in ZF-ZF, 75% in ZF-CF) (Fig. 2). Units with higher spontaneous rates were found in ZF-ZF units, but not in ZF-CF midbrain units. As high spontaneous rate units were infrequent in the ZF-ZF birds, our

Logerot et al.
2020
10.7717/peerj.9363

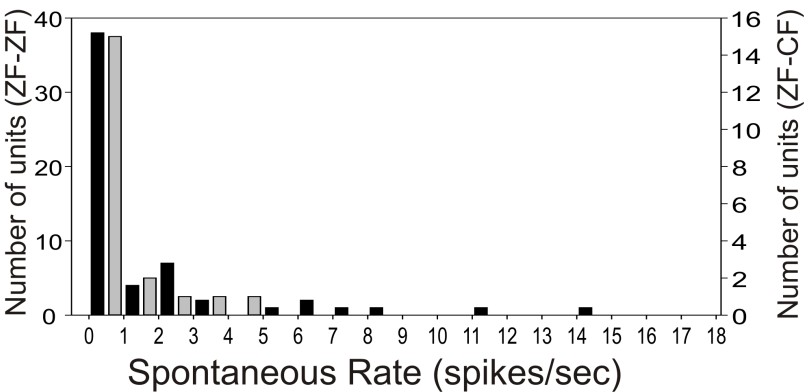

**Figure 2 Distribution of spontaneous rate of the units recorded in MLd.** Most units show spontaneous rates below 1 spike/sec. Black bars: ZF-ZF, grey bars: ZF-CF. Note difference in *Y* axis scale.

**Table 1 Percentage (number) of units responding to WN and individual tones in normal reared (ZF-ZF) and cross fostered (ZF-CF) birds.**

|        | WN        | 1 kHz     | 2 kHz     | 3 kHz     | 4 kHz     | 5 kHz    | 6 kHz   | None   |
|--------|-----------|-----------|-----------|-----------|-----------|----------|---------|--------|
| ZF-ZF  | 86% (50)  | 76% (44)  | 72% (42)  | 48% (28)  | 24% (14)  | 10% (6)  | 2% (1)  | 2% (1) |
| ZF-CF  | 80% (16)  | 75% (15)  | 85% (17)  | 75% (15)  | 15% (3)   | 30% (6)  | 5% (1)  | 0% (0) |

**Table 2 Percentage (number) of units with different BF in normal reared (ZF-ZF) and cross fostered (ZF-CF) birds.**

|        | 1 kHz     | 2 kHz     | 3 kHz     | 4 kHz    | 5 kHz    | 6 kHz   |
|--------|-----------|-----------|-----------|----------|----------|---------|
| ZF-ZF  | 26% (15)  | 34% (20)  | 24% (14)  | 5% (3)   | 2% (1)   | 0% (0)  |
| ZF-CF  | 35% (7)   | 30% (6)   | 25% (5)   | 5% (1)   | 5% (1)   | 0% (0)  |

failure to find units with strong discharge rates in cross fostered zebra finches may be related to the smaller number of units recorded.

Unit responses to white noise (WN) and tones are summarised in Table 1. Most ZF-ZF units and all ZF-CF units responded to at least one of the simple stimuli. Unit responses showed mostly excitation, although inhibition was also seen (6 units in ZF-ZF and 3 units in ZF-CF). Most units showed responses to WN and at least one of the tone stimuli, and most responded to more than one frequency, although 4 units in the ZF-ZF MLd failed to respond to any of the tones.

We derived the neurone's best frequency (BF) from the response curves of each unit when excited by the presentation of tones at 75 dB SPL. Most units were mainly tuned to lower frequencies, 2 kHz being the most represented best frequency, followed by 1 and 3 kHz (Table 2).

The temporal response patterns of each unit are based on a qualitative assessment of the shape of the PSTH obtained in response to BF and to WN and are summarised in Fig. 3E. Units were classified as onset, sustained, primary-like or primary-like with notch

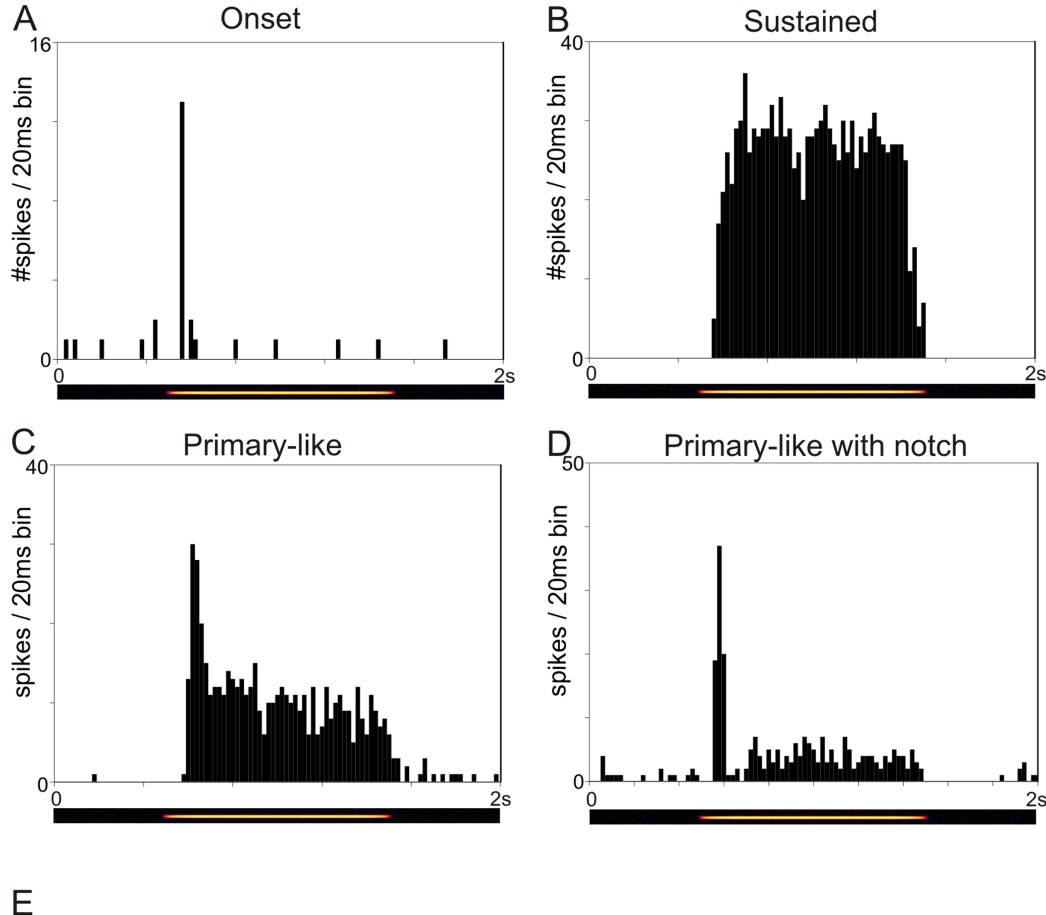

**Figure 3 Examples of the four temporal response patterns found in the normal zebra finch MLd.** All PSTHs in this figure have 20 ms bins on the time axis and represent response of units to their best frequency. The pure tone lasted 1 s and was preceded and followed by 500 ms of silence. (A) PSTH of a unit classified as onset showing with a typical strong response at the onset of the stimulus (yellow line in the underlying spectrogram) followed by either no or little response during the rest of the stimulation. (B) PSTH of a unit classified as sustained showing continued evoked response throughout the stimulus presentation. (C) PSTH of a unit classified as primary-like showing a strong response at the onset of the stimulus, followed by a continued but less vigorous firing rate during the duration of the rest of the stimulus. (D) PSTH of a unit classified as primary-like with notch, with a response pattern similar to that of the primary-like response but with a visible dip in firing rate shortly after stimulus onset. (E) Table showing the percentages (number) of units per temporal response patterns to their best frequency (BF) and to white noise (WN) stimulation in ZF-ZF and ZF-CF birds.

(Figs. 3A–3D). Not all units could be confidently classified into any of these 4 categories based on stimulation with BF or WN. Units showing a sustained response to BF and WN were the most frequently found in both ZF-ZF and ZF-CF.

**Table 3 Responses to different types of stimuli by MLd units in ZF-ZF (58 units) and ZF-CF (20 units).**

|  | ZF-ZF (%) | ZF-CF (%) |
|---|---|---|
| Responsive only to songs | 2 | 0 |
| Responsive to tones and songs | 12 | 25 |
| Responsive to WN and songs | 7 | 0 |
| Responsive to all 3 stimuli | 79 | 75 |
| Tones > WN | 52 | 20 |
| WN > Tones | 48 | 80 |

Tuning properties and response patterns to tones and WN of MLd units were comparable in the midbrains of ZF-ZF and ZF-CF birds. Thus, our qualitative analysis failed to detect an effect of rearing on the basic tuning properties of MLd neurones to simple stimuli beyond that which may be accounted for by differences in sample size.

## Effect of learning in responses to song

Behavioural data have shown that normal-reared zebra finches prefer to spatially associate with conspecific song over Bengalese finch song, whereas zebra finches that were cross fostered by Bengalese finches associated equally with both types of song. Using birds from the same population used for these behavioural experiments, we sought to examine the extent to which responses of auditory units in the auditory midbrain mirrored these behavioural differences. We challenged single units in the MLd of normal reared (ZF-ZF) and cross fostered (ZF-CF) zebra finches with 12 song stimuli: 3 conspecific (CON) songs and 3 heterospecific, Bengalese finch (HET) songs. All song stimuli were presented in the forward (FOR) and reverse (REV) direction. Of all the units that were classified as auditory in the present study most (ZF-ZF: 79%; ZF-CF: 75%) responded to all types of stimuli (WN, tones, and songs). A minority of neurones responded only to song stimuli (Table 3). Those units where WN failed to elicit a response did respond to both tones and song stimuli. Most units were excited by the stimuli, although inhibition was also seen. One unit in a ZF-CF bird was found to be inhibited by WN, 1 kHz, 2 kHz, and 3 kHz stimuli, but excited by 5 kHz and a few song stimuli.

Both conspecific (CON) and heterospecific (HET) song elicited robust responses from MLd units in ZF-ZF and ZF-CF zebra finches. Thirty-four of the 58 units of ZF-ZF (59%) and 15 of the 20 units of ZF-CF (75%) responded to all 12 song stimuli. The remaining 24 units (41%) did not respond to at least one of the song stimuli (Fig. 4). Only 10% of the ZF-ZF and 15% of the ZF-CF units responded to 8 songs or less. There was no indication of any bias towards conspecific song or forward version of the song at this level of analysis.

Response patterns of MLd units of ZF-ZF and ZF-CF birds, were heterogenous, with many units showing strong responses throughout the song stimulus presentation. Qualitatively, responses appeared to be mainly isomorphic, following the AM envelope of the conspecific or heterospecific stimulus (Fig. 5A) with fewer units responding in a more sustained manner throughout the entire song stimulus (Fig. 5B). Other unit responses

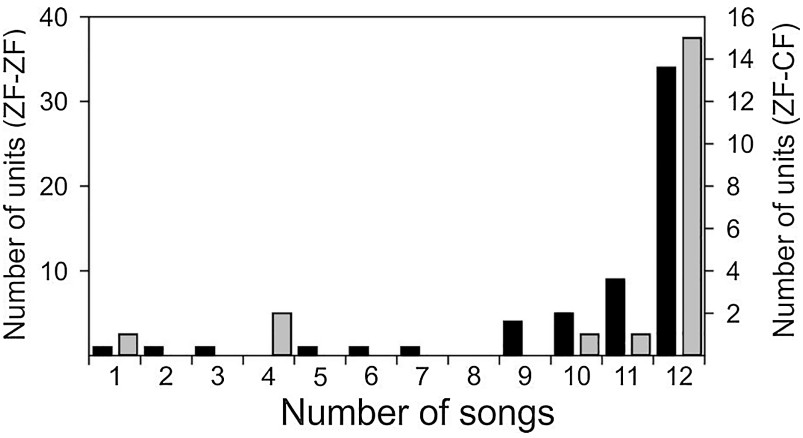

**Figure 4 Distribution of units of ZF-ZF (black) and ZF-CF (gray) responding to different number of songs.** In both normal reared (ZF-ZF) and cross-fostered (ZF-CF) zebra finches, most units responded to all 12 song stimuli, and in both groups, units were found to respond to a small number of the song stimuli.

appeared to show some degree of selectivity in their responses, either by responding more selectively to some elements within songs or by responding much more vigorously to some songs in particular (Figs. 5C and 5D).

The type of response to one stimulus was not, however, a good predictor of the unit's response to another one. For example Fig. 6 shows the responses of a ZF-CF MLd unit which is inhibited by white noise (Fig. 6A), and conspecific song 2 (CON-2) but increases its firing rate in response to heterospecific song 2 (HET-2) (See also Supplemental Table). Note that the increased response at the onset of CON-2 corresponds to a long silent period within the stimulus. Some other units would only respond to a subset of songs.

Determining what aspects of the song elicit these different types of responses is beyond the scope of this study. These results underscore the complexity of the neuronal responses encountered in this study, and show that the population of units in this study was not homogeneous with respect to response characteristics.

### d′ and SI

To compare the responses of units to pairs of stimuli we calculated the selectivity index (SI) and $d'$. In the present study, a $d'_{A \rightarrow B} > 2$ indicates a neurone's stronger response to stimulus $A$ over stimulus $B$ while a $d'_{A \rightarrow B} < -2$ indicates a neurone's stronger response to stimulus $B$ over stimulus $A$. A value of $d'$ falling between $[-2, 2]$ (non-selectivity zone) indicates no preference of one stimulus over the other (see "Material and Methods"). We plotted the cumulative distribution of the $d'$ values for a given song against its reverse version and the other 5 forward songs for all instances in normal reared birds (58 units × 5 −1 comparisons) and for all instances in cross-fostered zebra finches (20 units × 5 comparisons).

Figure 7 shows these distributions of $d'$ for all instances when conspecific song 2 (CON-2, in ZF-ZF MLd, Fig. 7A and ZF-CF MLd, Fig. 7D) and heterospecific song 1

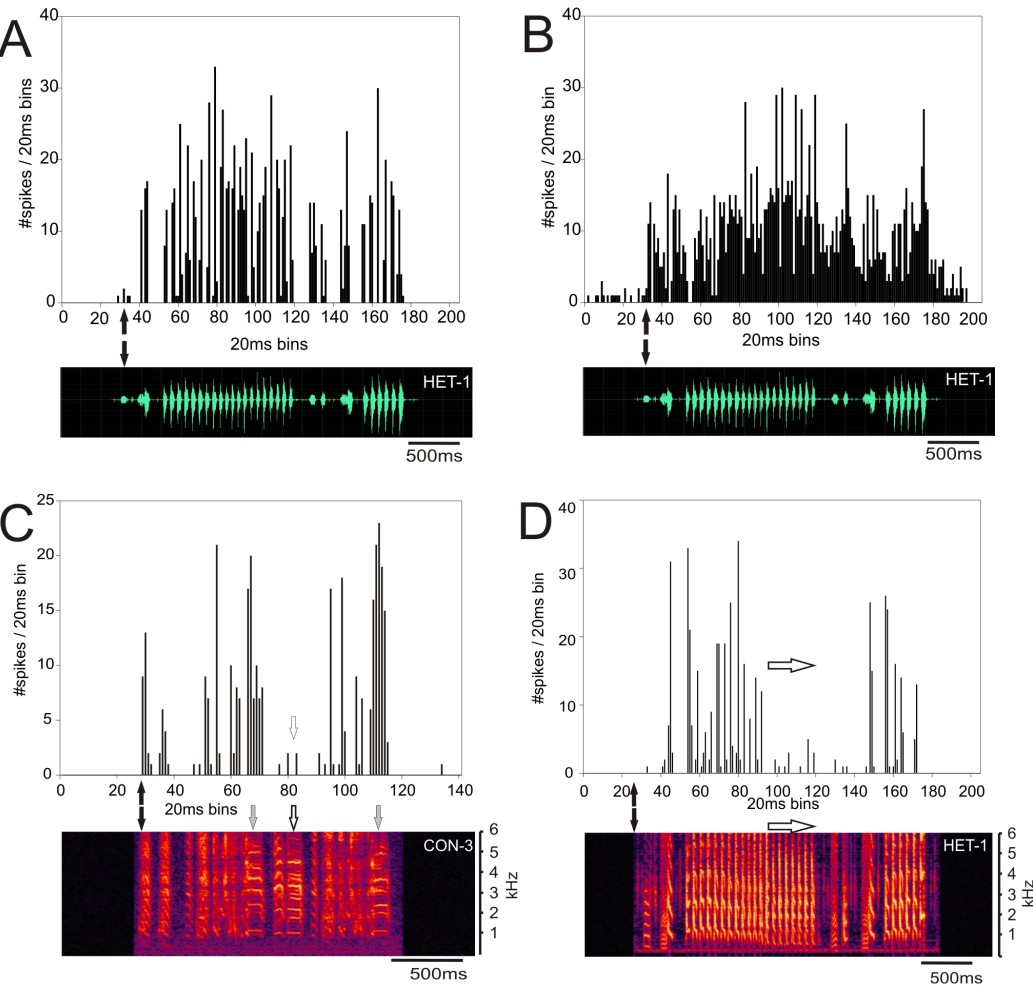

**Figure 5 Examples of response patterns to song in MLd units from normal reared zebra finches (ZF-ZF).** The black arrows indicate the beginning of the sound stimulus and of the unit's response. (A) Example of a unit where the responses are locked to the AM envelope of the song and responded to most, if not all the components of the song. (B) Example of a unit responding with a sustained pattern throughout the presentation of the song. (C and D) Examples of the same unit's response to a conspecific and heterospecific song. In response to conspecific song (C) the unit shows a strong responses to two but not all harmonic stacks (grey arrows and white arrows respectively). In response to a heterospecific song that does not contain harmonic stacks, the same unit shows strong responses to the first series of trills, but not to the second half of the trills (white arrows).

(HET-1, in ZF-ZF MLd, Fig. 7C and in ZF-CF MLd, Fig. 7B) were compared against their respective reverse version and all other forward versions of songs. Most comparisons fall within the non-selectivity interval, but many comparisons are found outside of this interval. In the case of CON-2, for those instances showing preference, most prefer CON-2 over its reverse or other forward songs ($d'$ values > 2). As in the ZF-ZF, in cross fostered zebra finches the cumulative distributions of $d'$ values of each song against all others for all instances suggest that, at the population level, there is no preference for an individual song stimulus with most comparisons falling within the non-selectivity zone (Fig. 7B).

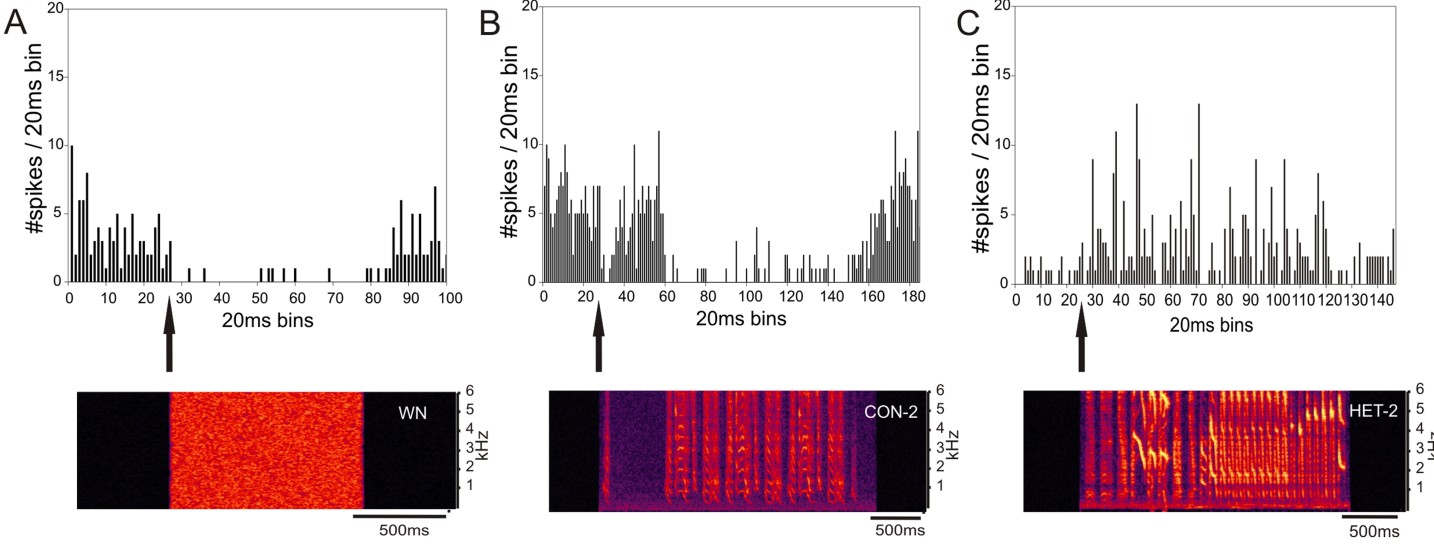

**Figure 6 Inhibition and excitation by different stimuli in the same MLd unit of a cross-fostered zebra finch (ZF-CF).** The black arrows indicate the beginning of the sound stimulus and of the unit's response. This unit showed inhibition in response to white noise (A) and a conspecific song (B), but is excited with the heterospecific song (C).

At the unit level, all but 2 units from ZF-ZF and 1 in ZF-CF MLds showed discrimination to at least one pair of stimuli ($d'$ outside the non-selectivity interval, Fig. 8). A unit's discrimination between a pair of stimuli could not predict discrimination response to a second pair.

Taken together, these results show that the majority of units in the MLd of both ZF-ZF and ZF-CF are able to discriminate at least one feature within the stimulus set.

### Response strength

The response strength (RS) measures the amount of activity within the total firing rate that can be directly attributed to the presentation of the stimulus. The RS to each song stimulus of the MLd units recorded in this study ranged from −3.58 to 41.37 spikes/s in ZF-ZF and −0.28 to 17.03 spikes/s in ZF-CF zebra finches, and there was no tail towards high values of RS in ZF-CF zebra finches (Fig. 9). In the ZF-ZF auditory midbrain, other than the peak at 0, the majority of instances were confined between 0 and 5 spikes/s whereas in ZF-CF higher RS values appear to fall less sharply (Fig. 9).

Thus, MLd units from cross-fostered zebra finches appeared to have lower RS values and a narrower range of RS values than those in normal reared birds, although formal statistical analysis indicated that the mean difference was not statistically significant ($F_{1,15} = 0.33$, $P = 0.58$). The RS values obtained for the forward and reverse presentation of a given song stimulus in a single unit are plotted against each other in Fig. 10. In 59% of the instances (here 58 units × 6 −1 FOR/REV song comparisons), slightly greater responses of the units toward the reverse version of songs were observed, which was confirmed by GLMM analysis ($F_{1,467} = 16.87$, $P = 0.0001$).

In order to investigate factors influencing the response strength of units within birds, a GLMM was fitted with bird and unit within bird as random effects to allow for the

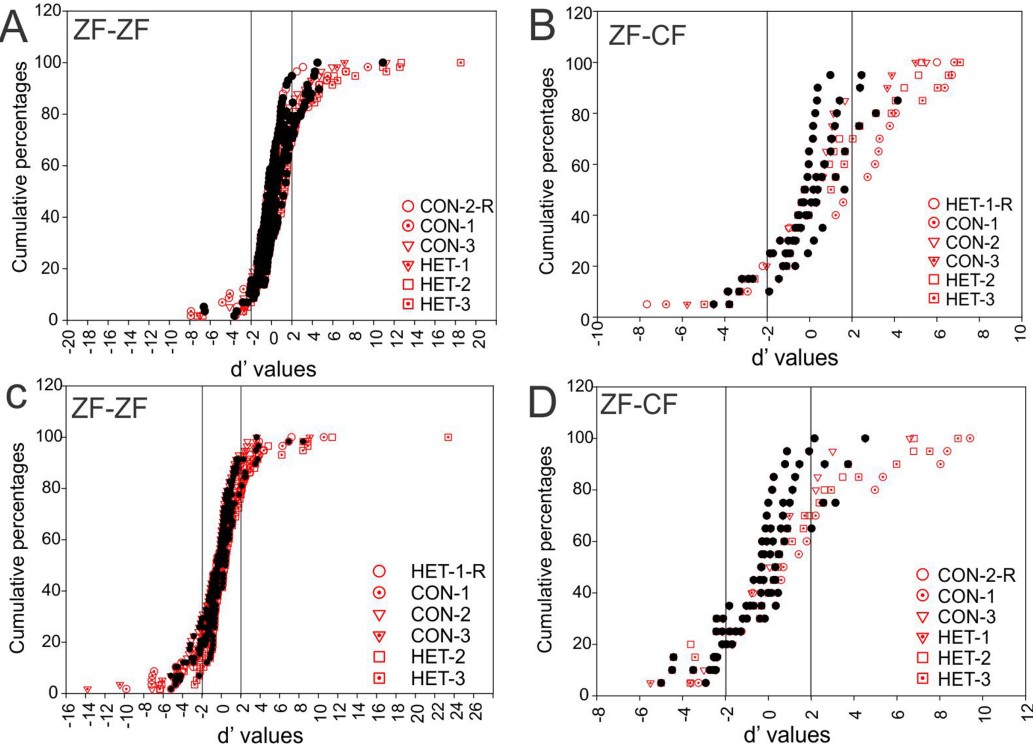

**Figure 7 Cumulative distribution of $d'$ values.** (A) Cumulative distribution of $d'$ values for the 58 individual MLd units recorded from in response to conspecific song 2 (CON-2) against its reverse version and all other forward song stimuli in normal reared zebra finches (ZF-ZF). (B) Cumulative distribution of $d'$ values for the 20 individual MLd units recorded from for heterospecific song 1 (HET-1) against its reverse version and all other forward song stimuli in cross-fostered zebra finches (ZF-CF). (C) Cumulative distribution of $d'$ values for the 58 individual MLd units recorded from in response to heterospecific song 1 (HET-1) against its reverse version and all other forward song stimuli in normal reared zebra finches (ZF-ZF). (D) Cumulative distribution of $d'$ values for the 20 individual MLd units recorded from for conspecific song 2 (CON-2) against its reverse version and all other forward song stimuli in cross-fostered zebra finches (ZF-CF). In all plots black-filled circles represent non-selective instances as determined by SI value and unfilled circles represent selective instances as determined by SI value (see "Methods") The vertical black line delimits the non-selectivity zone where $d' > |2|$.

clustering of units within birds, and song and direction (i.e. song played either forwards or backwards) as repeated measures with unstructured covariance. Bird Category (reared with conspecifics or cross-tutored), song type (conspecific or heterospecific, CON/HET), song within type (conspecific song 1, 2 or 3; heterospecific song 1, 2 or 3), direction of song (forward or reverse, FOR/REV) as well as the interactions of category, song type and song direction and the interaction of song with direction were initially included as explanatory variables and then interactions that were not significant were removed from the analysis (*Rouder et al., 2016*), as explained and justified in the Analysis section of the "Methods".

Initially, a full factorial model was fitted to the data. Using this model, no significant difference was found between ZF-ZF and ZF-CF birds as a main effect ($F_{1,15} = 0.43$, $P = 0.51$) but there were significant main effects of individual songs (Song ID)

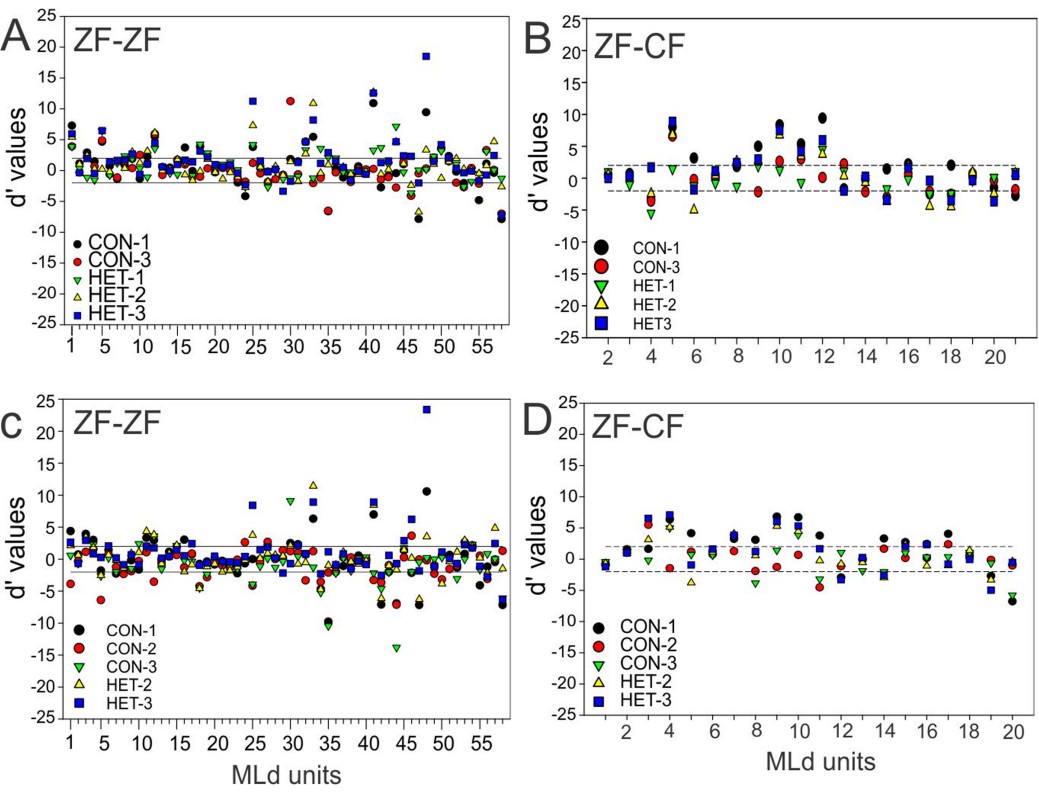

**Figure 8 Distribution of $d'$ values.** (A) Distribution of $d'$ values for 58 MLd units from normal reared zebra finches (ZF-ZF) for conspecific song 2 (CON-2) against other forward song stimuli. (B) Distribution of $d'$ values for 20 MLd units from cross-fostered zebra finches (ZF-CF) for conspecific song 2 (CON-2) against all other forward song stimuli. (C) Distribution of $d'$ values for 58 MLd units from normal reared zebra finches (ZF-ZF) for heterospecific song 1 (HET-1) against other forward song stimuli. (D) Distribution of $d'$ values for 20 MLd units from cross-fostered zebra finches (ZF-CF) for heterospecific song 1 (HET-1) against all other forward song stimuli. In all plots the horizontal black lines delimit the non-selectivity zone where $d' > |2|$. Data points above the non-selectivity zone indicate instances in which CON-2 (A and B) or HET-1 (C and D) were preferred over the other songs they were tested against. Data points below the non-selectivity zone indicate instances in which the other song stimuli were preferred over CON-2 (A and B) or HET-1 (C and D).

($F_{4,384}$) = 4.4, $P = 0.003$) and the direction of songs (FOR/REV: $F_{1,467} = 13.26$, $P = 0.0001$). A reduced model was investigated and compared to the initial model using the AIC values, where the smaller value is better. When the reduced model was used, on the basis of the smaller AIC value (*Rouder et al., 2016*), the results were, in general, similar: Bird Category as a main effect was non-significant ($F_{1,15} = 0.33$, $P = 0.58$), individual song (Song ID) as a main effect was significant ($F_{4,384} = 4.95$, $P = 0.0007$), as was FOR/REV ($F_{1,467} = 16.87$, $P = 0.0001$). In addition, there was a significant difference between ZF-ZF and ZF-CF birds in the effect of CON or HET, with ZF-ZF birds having larger reductions in response to HET than to CON compared to ZF-CF birds (i.e. a significant interaction: ($F_{1,384} = 4.12$, $P = 0.043$; see Tables 4 and 5; Figs. 9B and 10). These appear to result from a reduction in response strength to conspecific song and an increase in response strength to the song of the foster species (mean RS: ZF-ZF = 5.76 (CON), 4.49 (HET); ZF-CF = 5.20

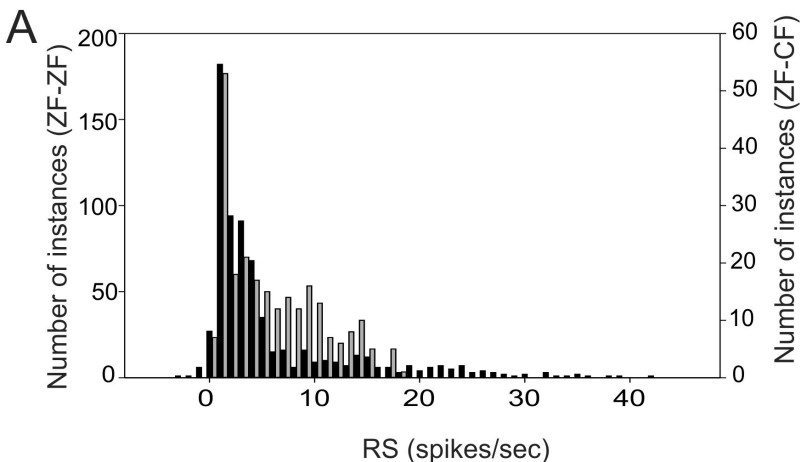

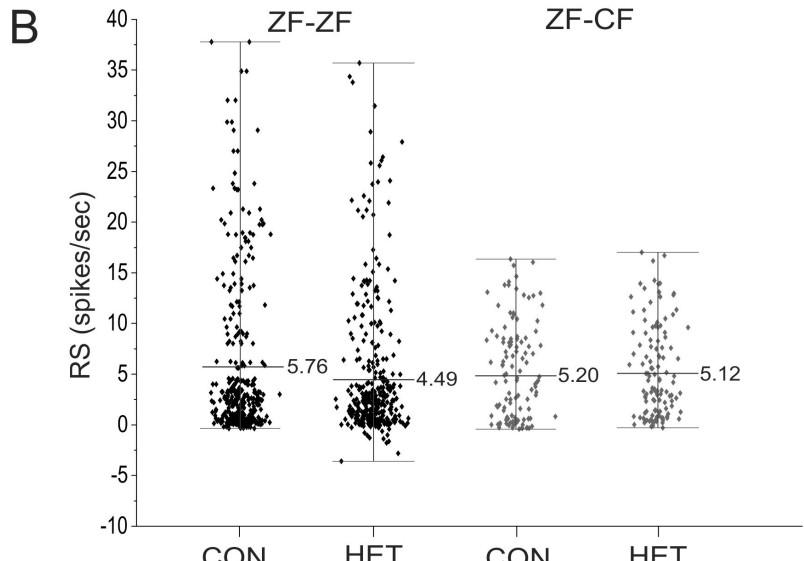

**Figure 9 Distribution of RS values.** (A) Distribution of RS values (spikes/s) for all song stimuli presentations from MLd units from normal reared (ZF-ZF, black bars) and cross-fostered (ZF-CF, grey bars) zebra finches. (B) RS (spikes/s) values for ZF-ZF (black) and ZF-CF (gray) for all units in response to conspecific (CON) or heterospecific (HET) stimuli. The range and mean of the responses are indicated for each group.              

(CON), 5.12 (HET); both $P = 0.0001$; see Table 5). CON/HET was also significant as a main effect but because it was significant in interaction with Bird Category, it was not examined further as an independent effect (*Nelder, 1977*; *Martinez, 2015*). Despite this strong evidence that direction influences RS depending on rearing, the size of the effect was not large. This is demonstrated when the RS values obtained for the forward and reverse presentations of a given song stimulus in a single unit are plotted against each other where most data points fall on or near the equality line (Fig. 10).

  These results show that the MLd units of all birds did not respond equally to individual songs within one category (CON or HET) and indicate that early auditory experience shapes responses in the auditory midbrain in the zebra finch (Table 4).

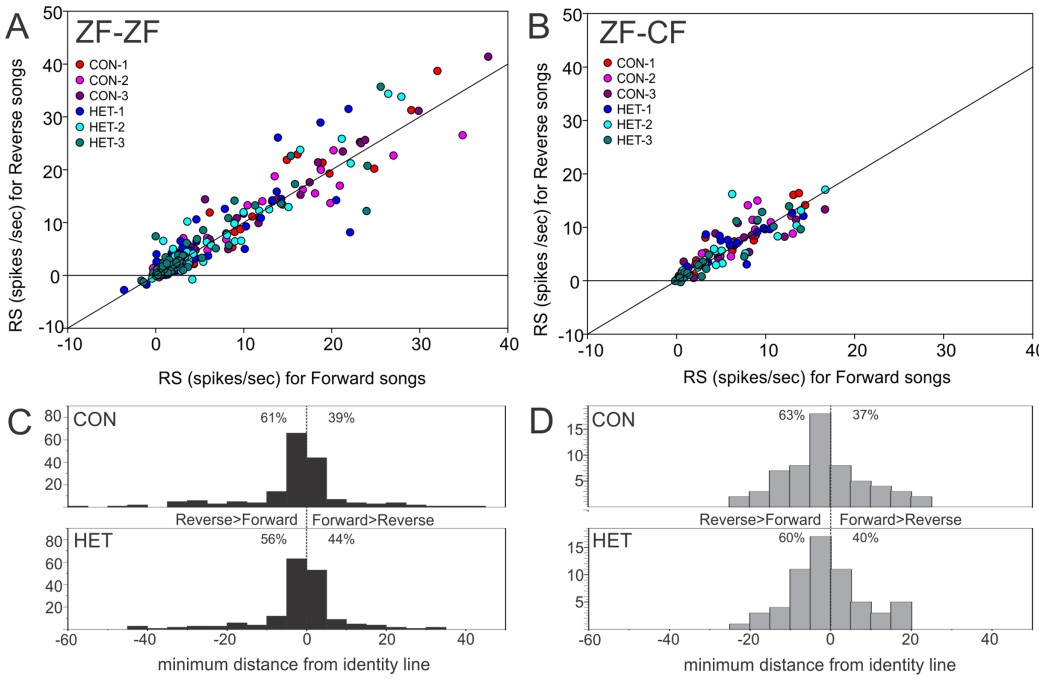

**Figure 10 Mean RS value of individual units.** Plots of mean RS values for each individual unit in response to a forward song vs. its reverse version. (A) Normal reared zebra finches (ZF-ZF); (B) cross-fostered zebra finches (ZF-CF). In both plots the diagonal line indicates equal response to forward and reverse version. (C) and (D) Histograms showing the minimum distance of individual data points to the identity line calculated as: (RS to FOR version of the song + RS to REV version of the song)/√2. Positive values were assigned to instances where response to FOR song is larger than to REV song and negative values were assigned to instances where responses to REV song are larger than to FOR song. Percentages indicate the proportion of units that fall either above, or below the identity line.

**Table 4 Significant main effects and interactions as determined by the GLMM analysis.**

| Effect | Num DF | Den DF | *F* Value | Pr > *F* |
|---|---|---|---|---|
| Type 3 tests of fixed effects | | | | |
| Individual song | 4 | 384 | 4.95 | 0.0007 |
| Song direction | 1 | 467 | 16.87 | <0.0001 |
| Rearing × CON/HET | 1 | 384 | 4.12 | 0.0429 |

## DISCUSSION

We sought to describe the responses of MLd units in normal reared zebra finches (ZF-ZF) that sing a zebra-finch typical song, and zebra finches that had learned an abnormal song through cross fostering with Bengalese finches (ZF-CF). MLd units in both ZF-ZF and ZF-CF showed comparable responses to white noise and tones. In contrast, birds that were cross-fostered modified the responses of MLd units to conspecific and heterospecific song stimuli in a way that parallels behavioural shifts in song preference in the same population of birds as that used by *Campbell & Hauber (2009, 2010)*. We therefore

**Table 5 Least squares means.**

| Effect | Category | Song ID | Song type | Direction | Estimate | Standard error | DF | T value | Pr > \|t\| |
|---|---|---|---|---|---|---|---|---|---|
| Least squares means | | | | | | | | | |
| Category × song type | ZF-CF | | CON/HET | | 0.9706 | 0.05651 | 22.7 | 17.18 | <0.0001 |
| Category × song type | ZF-ZF | | CON/HET | | 0.9479 | 0.04053 | 8.3 | 23.38 | <0.0001 |
| Direction | | | | FOR/REV | 0.9407 | 0.03458 | 15 | 27.21 | <0.0001 |
| Song (song type) | | CON-1 | CON | | 0.9380 | 0.03551 | 16.8 | 26.41 | <0.0001 |
| Song (song type) | | CON-2 | CON | | 0.9679 | 0.03551 | 16.8 | 27.25 | <0.0001 |
| Song (song type) | | CON-3 | CON | | 0.9719 | 0.03551 | 16.8 | 27.37 | <0.0001 |
| Song (song type) | | HET-1 | HET | | 0.9506 | 0.03551 | 16.8 | 26.77 | <0.0001 |
| Song (song type) | | HET-2 | HET | | 0.9466 | 0.03551 | 16.8 | 26.65 | <0.0001 |
| Song (song type) | | HET-3 | HET | | 0.9128 | 0.03551 | 16.8 | 25.70 | <0.0001 |

conclude that the effects of developmental experience, although not limited to hearing and learning non-conspecific vocalizations during development, but which remove the preference for conspecific song in behavioural experiments, are manifested in the response properties of brainstem auditory neurones.

## General description

### Spontaneous rate

In agreement with previous studies (*Woolley & Casseday, 2004*) most of our MLd units showed no or low spontaneous activity. *Woolley & Casseday (2004)* reported 90% of their units had no spontaneous activity, with an upper limit of 7 spikes/s. In contrast, we found that units with no spontaneous activity were less common (22% in ZF-ZF and 20% in ZF-CF), and some units in ZF-ZF MLd had spontaneous rates to 14–15 spikes/s. These discrepancies are not surprising given that Woolley and Casseday used urethane as an anaesthetic in contrast to ketamine/xylazine in this study. When compared to unanaesthetized birds, *Schumacher, Schneider & Woolley (2011)* showed that MLd units in urethane anaesthetized male zebra finches had lower spontaneous activity than in unanaesthetized birds. Similarly, in the inferior colliculus of the guinea pig, *Astl et al. (1996)* found lower rates of spontaneous activity under urethane anaesthesia than under ketamine anaesthesia. Thus, discrepancies in the range of spontaneous activity are likely attributable to the anaesthetic regime. We did observe, however, that MLd units from ZF-CF finches showed a narrower range of spontaneous activity than those from ZF-ZF finches. *Woolley, Hauber & Theunissen (2010)* do not report if cross-fostering results in changes in spontaneous activity, although they do report that cross-fostering reduces the firing rate of MLd units. Since in normal-reared zebra finches units with high spontaneous activity are infrequent, it is possible that these differences are the result of a smaller sample size.

### Response characteristics

Most of the units we recorded from responded to all types of stimuli (WN, tones, song: 79% of ZF-ZF units and 75% of ZF-CF units, Table 3), the remainder responding to a

subset of the stimuli. Consistent with previous work, we found several temporal response patterns in MLd units from ZF-ZF to ZF-CF finches when stimulated with best frequency (BF) and white noise (WN) (*Woolley & Casseday, 2004*; *Schumacher, Schneider & Woolley, 2011*). Fewer units could be classified into a given temporal response pattern when stimulated with WN than with BF, but the proportion of units that could be classified into any one temporal response pattern was similar in ZF-ZF and ZF-CF. The most frequent temporal response pattern in both bird groups was the sustained type, followed by the onset type. The literature reports varying proportions of temporal response patterns. *Schumacher, Schneider & Woolley (2011)* found the primary-like response to be the most frequently found in the MLds of unanaesthetized and urethane anaesthetized zebra finches. In contrast, *Woolley & Casseday (2004)* found in urethane anaesthetized zebra finches that onset responses were the most frequent. The discrepancies between these two studies suggest that the differences in proportions of response types may be due to sampling biases between different studies. For example, in the inferior colliculus (IC) of the cat, the onset pattern was reported as the most common type by *Rose et al. (1963)*, whereas *Aitkin, Tran & Syka (1994)* report that the sustained units predominate over onset units. Thus, while providing descriptive value of the population at hand, differences in the proportion of response patterns may be difficult to interpret meaningfully.

Zebra finch and Bengalese finch songs have a comparable frequency range and both have harmonic stacks in their song, but diverge in other spectro-temporal elements. Zebra finch songs usually contain noisy and harmonic elements whereas Bengalese songs contain fast repeated harmonic syllables called trills (*Zann, 1996*; *Woolley & Casseday, 2004*). Zebra finches cross-fostered with Bengalese finch parents learn a song that contains BF elements within temporal gaps typical of zebra finch songs (*Araki, Bandi & Yazaki-Sugiyama, 2016*). Both zebra finch (CON) and Bengalese finch (HET) songs elicited robust isomorphic and non-isomorphic responses, but the response to one stimulus was not necessarily a good predictor of the unit's response to another. For example, frequency tuning alone could not explain the selective response to complex vocalizations. This mirrors results obtained in guinea fowl (*Scheich, Langner & Koch, 1977*), zebra finches (*Woolley, Gill & Theunissen, 2006*; *Schneider & Woolley, 2011*), bats (*Portfors, 2004*), and fish (*Maruska & Tricas, 2009*). Although our study was not designed to identify the precise features that evoked the unit's response, some degree of feature extraction in MLd was evident. Single units showed discrimination between the varied song types, although not all songs were preferred equally by MLd units, and no single song was consistently preferred. Some units responded to all songs and all elements within the songs, others selectively responded to subsets of songs or to particular spectro-temporal features within notes or syllables, whereas others were more tightly tuned to the amplitude modulations. *Woolley & Casseday (2005)* report that 79% of the units they recorded from in the male zebra finch MLd responded to amplitude modulated tones. *Woolley et al. (2009)* using spectral-temporal receptive fields (STRFs) to analyse the responses of zebra finch midbrain neurones to conspecific songs, identified four functional groups, each with the potential to contribute to feature extraction of the conspecific signals. More than half were broadband neurones whose characteristic STRF

made them either good at detecting onsets and suitable for encoding rhythm, or good at encoding amplitude envelope and suitable to participate in encoding of timbre. Distinct populations dedicated to encoding either spectral features or temporal information have also been described in fish (for review see *Bass & McKibben, 2003*) amphibians (for review, see *Feng, Hall & Gooler, 1990*; *Bass, Rose & Pritz, 2005*; *Rose & Gooler, 2007*), and mammals (for review see *Ehret & Schreiner, 2005*; *Rees & Langner, 2005*). In the present study we were able to detect subpopulations within MLd that could potentially contribute to the discrimination of conspecific vocalisations, providing further evidence that there is some degree of feature extraction at the level of MLd. Furthermore, the $d'$ and SI data suggest that cross fostering does not influence the ability of units to discriminate between natural stimuli. The heterogeneity of responses to conspecific stimuli in the auditory midbrain, together with a key role of inhibition in shaping the selectivity of neurones, has been described; lifting inhibition leads to a broader tuning curve and a loosening of the neurone's selectivity to complex vocalizations (*Yang, Pollak & Resler, 1992*; *Hall, 1999*; *Klug et al., 2002*; *Portfors, 2004*; *Xie, Meitzen & Pollak, 2005*; *Pollak, 2011*; *Mayko, Roberts & Portfors, 2012*). In zebra finches *Woolley & Portfors (2013)* showed that blocking GABA and glycine inhibition results in an expansion of the tuning curves of IC neurones and reduces the selectivity to vocalizations. It will be interesting to determine whether the reduced discrimination between CON-HET categories that is seen as a result of cross-fostering is associated with changes in GABAergic function in the MLd.

## Effect of rearing in responses to song

We sought to answer two questions: Do units in MLd show a response bias towards conspecific songs, and if this exists, can this be modified by developmental experience? To this end, we challenged MLd units of ZF-ZF and ZF-CF zebra finches with stimuli consisting of 3 familiar zebra finch (CON) and 3 unfamiliar Bengalese finch (HET) songs, each played forwards and backwards. We found that in both groups of birds, responses to song stimuli were heterogeneous, that units in MLd were able to discriminate between different stimuli, and that a preference towards conspecific song was developmentally modified.

The response parameter of choice for the GLMM analysis was RS because it better reflects the responses of the units to different song stimuli. While the SI is good at detecting selectivity to certain features, its shortcoming lies in its failure to detect the strength of that response. The RS, although it provides little information regarding the temporal pattern of the evoked response, is a more direct measure of the 'total' response of the cell. The distribution of RS values in ZF-CF differs from that seen with ZF-ZF units (Mann–Whitney $U$ test for equality of distributions, $P = 0.001$), although the mean differences were non-significant. *Woolley, Hauber & Theunissen (2010)* showed that cross-fostering resulted in a lower firing rate in MLd neurones. In contrast we find that the median RS for cross fostered birds is higher than for normal-reared birds (ZF-ZF: 2.4, ZF-CF: 4.4) (Mann–Whitney $U$ test for medians, $P = 0.0001$). *Schumacher, Schneider & Woolley (2011)* found that units in MLd of male zebra finches under urethane anaesthesia had lower spontaneous and song-evoked firing rates and response strength (compared

to unanaesthetised birds). In guinea pig IC, *Astl et al. (1996)* showed that frequency tuning was similar under urethane versus Ketamine/Xylazine anaesthesia, although the number of spontaneously active neurones was higher under Ketamine/Xylazine. While the anaesthetic regime may account for differences in overall response levels between our and Woolley's results, it is unlikely to account for the direction of the changes observed as a result of cross-fostering.

Three main findings resulted from the GLMM analysis of RS values. First, the reverse versions of songs evoked stronger responses than the forward versions of songs in both ZF-ZF and ZF-CF zebra finches. Second, there was an effect of individual songs within CON and HET categories. Third, ZF-ZF MLd units responded more strongly to CON than to HET songs than did their ZF-CF counterparts. Each of these findings will be considered separately below.

### Stronger response to reverse song

The stronger responses toward reverse songs is puzzling. *Woolley & Casseday (2005)* found that in MLd, 89% of the units they tested were responsive to FM sweeps, but most of these cells were insensitive to the direction of the sweep. Of the five cells that showed a preference, four preferred the upward sweep direction. Since the zebra finch song contains more downward FM than upward FM, a preference for upward sweeps would be expected to evoke a stronger response to reverse song in FM-sensitive neurones. The ICs of bats are known to be dominated by direction-selective neurones that favour the downward direction (*Suga, 1965*; *Fuzessery & Hall, 1996*; *Casseday, Covey & Grothe, 1997*; *Razak & Fuzessery, 2006*; *Andoni, Li & Pollak, 2007*; *Voytenko & Galazyuk, 2007*) and express a range of preferences for sweep velocities that correspond to the sweep velocities in the signals they emit (*Andoni, Li & Pollak, 2007*). However, and in contrast to our neural data, *Lauay et al. (2004)* found no significant difference in a choice task in the preference for forward song over reverse song in zebra finch females, whether they had been reared with both parents or isolated from male tutors. A preference for forward and reverse version of song in cross-fostered zebra finches has, to our knowledge not been examined.

### Individual song effect within CON-HET categories

Our results show that the MLd units of both ZF-ZF and ZF-CF birds did not respond equally to individual songs within one category (CON or HET). At the population level, not all songs were preferred equally, and no single song was consistently preferred. However, single-units were able to discriminate between the varied song types.
The heterogeneity of responses in MLd appears to include a large population of neurones that represent a large range of stimulus characteristics, with units showing highly reproducible responses that indicate a high degree of selectivity to stimulus parameters. More interestingly, the analysis also showed that there was an effect of song within either the conspecific or heterospecific signals. Thus, the choice of stimuli used for these kinds of studies, and whether they are considered an independent variable, might have an impact on the interpretation of the results.

### Effect of rearing in the preference to conspecific song

Selectivity toward the forward conspecific over the heterospecific songs in the MLd of ZF-ZF and ZF-CF birds was tested by *Woolley, Hauber & Theunissen (2010)*. Contrary to the results presented here, they did not find a difference in the responses to conspecific and heterospecific songs in male ZF-ZF or ZF-CF, although the mean firing rates to both types of stimuli were lower in the cross-fostered birds. In contrast, our study shows that response differences between CON/HET categories is reduced as a result of rearing experience, likely due to a reduction in response strength to conspecific song and an increase in response strength to the song of the foster species. Differences in the cross-fostering paradigm, the choice of songs as stimuli, or effects of the use of different anaesthetics may account for the discrepancies in the two studies. However, our neural data mirrors the behavioural preference reported in the same population of birds (*Campbell & Hauber, 2009*, *2010*). More likely, the discrepancies may result from the regions of MLd that were sampled. Although *Woolley, Hauber & Theunissen (2010)* do not report on the location of their recordings, the BF of the units in our study suggest our recordings were confined to more dorsal aspects of MLd (*Scheich, Langner & Koch, 1977*; *Coles & Aitkin, 1979*; *Calford, Wise & Pettigrew, 1985*; *Takahashi & Konishi, 1988*; *Woolley & Casseday, 2004*). In the guinea fowl *Scheich, Langner & Koch (1977)* report that recording from ventral MLd yielded units whose responses to calls were predictable from the unit's response to tones, whereas recordings from posterior-dorsal areas yielded a higher proportion of units whose response to complex calls could not be predicted from the neurone's tuning curve. Regional differences in the response to complex stimuli have also been reported in the IC of cat (*Aitkin, Tran & Syka, 1994*) and guinea pigs (*Lyzwa, Herrmann & Wörgötter, 2016*) and in the torus of frogs (*Hoke et al., 2004*; *Mangiamele & Burmeister, 2011*).

The importance of rearing environment in the development of preference for communication sounds in birds is well known. Swamp sparrows, for example show a behavioural preference for conspecific over heterospecific signals (*Dooling & Searcy, 1980*). Adult male and female zebra finches will prefer conspecific over heterospecific songs when challenged with a behavioural choice task, regardless of whether they have been raised in isolation from a male tutor or with both conspecific parents (*Braaten & Reynolds, 1999*; *Lauay et al., 2004*). Juvenile male and female zebra finches that were reared in acoustic isolation preferred conspecific song over heterospecific song (*Braaten & Reynolds, 1999*). Females raised with an adult male preferred the songs of tutored males over those of untutored males, while females that were deprived of the presence of a male did not show this preference (*Lauay et al., 2004*). Normally-reared birds, exposed solely to conspecifc vocalizations, respond preferentially to conspecific songs. But when deprived of the conspecific exposure and instead develop hearing a Bengalese finch song, zebra finches present similar responses to the songs of both their own species and those of the cross-fostering species.

## Ascending and descending inputs into MLd

We did not establish whether the responses we observed are influenced by ascending or descending inputs to MLd (*Pannese, Grandjean & Frühholz, 2015*). In mammals,

subcortical structures have been shown to contribute to the processing of vocalizations (*Bauer, Klug & Pollak, 2002*; *Pannese, Grandjean & Frühholz, 2015*; *Roberts & Portfors, 2015*), but recordings from the lower brainstem in songbirds have been limited to studies of tonotopy (*Konishi, 1969*; *Konishi, 1970*; *Sachs & Sinnott, 1978*). To our knowledge, no attempts have been made to examine the responses of hindbrain auditory nuclei in the processing of complex sounds to infer what influence they may have on MLd responses. It should be noted, however, that the terminal fields of ascending inputs to MLd originating in the hindbrain do not conform to the typical avian pattern where the nucleus laminaris (NL) terminal field defines an MLd core (*Conlee & Parks, 1986*; *Takahashi & Konishi, 1988*). Instead, the terminal fields appear to overlap more broadly within MLd (*Krützfeldt et al., 2010*). The organisation of the NL of songbirds also shows morphological modifications, the significance of which remains unknown (*Kubke & Carr, 2006*). Whether these anatomical differences reflect an adaptation for specific aspects of sound processing has not been established.

Descending inputs to the auditory midbrain have been shown to influence neuronal responses in a number of vertebrates (*Endepols & Walkowiak, 1999*; *Gao & Suga, 2000*; *Suga et al., 2000*; *Endepols & Walkowiak, 2001*; *Ma & Suga, 2001*; *Suga et al., 2002*; *Popelář et al., 2016*). It is thus possible that some of the changes observed in MLd may be influenced by forebrain auditory nuclei (*Gentner & Margoliash, 2003*; *Cousillas et al., 2004*; *Jeanne et al., 2011*; *George & Cousillas, 2013*; *Araki, Bandi & Yazaki-Sugiyama, 2016*). The auditory forebrain has been implicated in the categorisation of vocalizations (*Chew, Vicario & Nottebohm, 1996*; *Grace et al., 2003*; *Elie & Theunissen, 2015*). Strong responses to conspecific song have been recorded in Field L, and the development of neuronal selectivity for conspecific song appears to parallel the emergence of behavioural song preference (*Amin, Doupe & Theunissen, 2007*). Changes in forebrain responses that are influenced by development have also been reported in the ventral caudal hyperstriatum (cHV) (*Grace et al., 2003*) and caudal medial nidopallium (*Stripling, Kruse & Clayton, 2001*; *Menardy et al., 2012*). However, descending inputs to MLd in songbirds are not substantial. The only descending input that has been described is from the robust nucleus of the arcopallium (RA) cup (*Mello et al., 1998*), but these inputs are largely confined to a region external to MLd, and the adjacent intercollicular complex. Similarly with respect to the RA cup's projection to the vicinity of lower auditory brainstem nuclei, such as the dorsal and ventral nuclei of the lateral lemniscus (LLD and LLV), both of which project strongly to MLd (*Wild, Krützfeldt & Kubke, 2010*). But in neither the case of MLd nor in that of LLD and LLV have injections of neural tracers been found to retrogradely label adjacent neurones in the region of the RA cup's descending projections.

## CONCLUSION

One interesting observation in our study is that the differences in responses to conspecific and heterospecific songs between ZF-ZF and ZF-CF finches were present even though both groups of birds were housed in the same aviary as adults, thus exposed to the same sound environment. This suggests that the differences in response to conspecific and heterospecific songs recorded in MLd are not simply a result of repeated exposure to a

particular auditory stimulus. Repeated exposure has been shown to change neural responses in the auditory cortex of rats (*Bao et al., 2013*), ferrets (*Schnupp et al., 2006*), and gerbils (*Caras & Sanes, 2017*), and in the auditory midbrain of mice (*Cruces-Solís et al., 2018*) and frogs (*Gall & Wilczynski, 2014*). Instead, the persistence of the differences observed in the MLd of both groups of birds, suggest that these changes may be influenced by the critical period of learning associated with song learning. A 'memory' of learned song in the MLd of zebra finches has been suggested by *Van der Kant et al. (2013)*.

Our results further confirm that neurones in MLd are tuned to characteristics of the conspecific song that allow discrimination between conspecific and heterospecific *learned* vocalisations. These neuronal 'filters' can be modified during the period of song learning in the direction that would support the learning of a heterospecific song. Further studies in juvenile birds will be needed to determine the developmental trajectory of these changes, and how that trajectory maps onto changes in ascending and descending inputs and the emergence of behavioural preference for conspecific song.

## ACKNOWLEDGEMENTS

This work was previously published as a PhD thesis (*Logerot, 2011*). We would like to thank Dana Campbell for providing the birds used in this study, and Joanna Stewart for initial assistance with statistical analysis.

### Funding

This work was supported by the Marsden Fund (administered by the Royal Society of New Zealand). The funders had no role in study design, data collection and analysis, decision to publish, or preparation of the manuscript.

### Grant Disclosures

The following grant information was disclosed by the authors:
Marsden Fund, Royal Society of New Zealand.

### Competing Interests

M. Fabiana Kubke is an Academic Editor for PeerJ.

### Author Contributions

- Priscilla Logerot conceived and designed the experiments, performed the experiments, analysed the data, prepared figures and/or tables, authored or reviewed drafts of the paper, and approved the final draft.
- Paul F. Smith analysed the data, prepared figures and/or tables, authored or reviewed drafts of the paper, and approved the final draft.
- Martin Wild conceived and designed the experiments, analysed the data, prepared figures and/or tables, authored or reviewed drafts of the paper, and approved the final draft.

- M. Fabiana Kubke conceived and designed the experiments, analysed the data, prepared figures and/or tables, authored or reviewed drafts of the paper, and approved the final draft.

## Animal Ethics

The following information was supplied relating to ethical approvals (i.e., approving body and any reference numbers):

University of Auckland Animal Ethics Committee in accordance with the University of Auckland Code of Ethical Conduct for the Use of Animals for Teaching and Research, the Animal Welfare Act 1999 (New Zealand), and The National Animal Ethics Advisory Committee (NAEAC) Good Practice Guide for the Use of Animals in Research, Testing and Teaching, approved this research (R425).

## Data Availability

The raw data are available at Figshare: Kubke, M. Fabiana; Logerot, Priscilla; Wild, Martin; Smith, Paul F. (2020): MLd responses in normal reared and cross fostered zebra finches. figshare. Dataset. DOI 10.6084/m9.figshare.11828958.v2.

## Supplemental Information

Supplemental information for this article can be found online at http://dx.doi.org/10.7717/peerj.9363#supplemental-information.

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
