# Peer review of "Auditory processing in the zebra finch midbrain: single unit responses and effect of rearing experience"

_PeerJ, doi:10.7717/peerj.9363_

## Round 0.1 · original submission · Major Revisions

Please address both reviewers' comments point by point. In particular, there are concerns by both reviewers regrading your data analysis and the validity of your conclusions.

Reviewer 1 ·

Basic reporting

This paper was well written and clear. I have only a few suggestions that I think should aid clarity.

With regard to the methods section.

Currently, the measure or Response strength is after the discussion of Selectivity index and D prime though the Response strength is used in the calculation of Selectivity index and D prime. This organization mirrors that in the results section. I think, in the Methods section, Response strength should come before Selectivity and D prime so that the meaning of those calculations will be clearer to the reader.

Currently, there is no comprehensive description of the terms in the equations provided. This can lead to some confusion in reading the equations. For example, in the calculation of D prime, there is a term: sigma^2 RSa. This could be interpreted as, the variance of the distribution of the Response strength to stimulus A, or (as RSa is previously used for the Response strength (which is a mean value) to stimulus A), it could be interpreted as some variable, sigma^2 times the Response strength to stimulus A. A single sentence following each equation defining terms would resolve any confusion.

The methods section covering the GLMM is incomplete. There should be an equation presented for the GLMM. It is confusing to try to reconstruct this equation from the text as written and a formal equation for the model that they are using would help dramatically. Absent that, a pseudo-formula (perhaps that used in the software package to fit the model) would help clarify dramatically.
Further, this section needs to describe the algorithm used to fit variable weights. The method used can impact the results. There is generally no analytical solution to GLMMs and different approaches to fitting the model will impact the results. The analysis is not complete without clarity on how the model was fit. Additionally, the software package used to fit the GLMM should be indicated and a reference provided.
Finally, all of the coefficients and their variances (where calculated) should be reported, not just the significant coefficients. It is difficult to interpret the meaning of specific coefficients without information about all coefficients in a model especially if there are significant interaction terms as there are here.

Throughout the first portion of the paper individual neurons are categorized into distinct classes, but, there is no statement of how those classes were defined or how individual neurons were categorized. There should be a section in the Methods describing both in some detail.

Experimental design

I think the experimental design is sound.

Validity of the findings

The major conclusion of this paper is that different life experiences (and probably exposure to different tutor songs) results in different tuning of neurons in lower (MLd) auditory areas. This conclusion hinges on two main experimental results. First, that birds (Zebra finches) who are cross fostered to parents of a different species (Bengalese finches) have an overall decrease in neural responsiveness (in MLd) to the auditory stimuli presented when compared to a cohort of birds reared by members of their same species. Second, if you model the responsiveness of neurons in MLd to different stimuli using a GLMM you find that an interaction between rearing condition (reared by same species vs reared by a different species) and whether the song was heterospecific or conspecific.

For the first result to be believable, there must be a statistical test demonstrating that responsiveness really has changed. From looking at the distributions in figure 9 it is likely that a statistical test will show a change between those distributions, but a test should be done. If no parametric or non parametric test seems sensible, a monte-carlo re sampling test could be used.

For the second result to be believable all of the weights found in the process of modeling should be reported. It is very difficult to interpret the meaning of a specific weight without information about the other fitted weights especially when one of the parameters is an interaction effect. All the weights should be provided along with the other methodological details asked for above.

Reviewer 2 ·

Basic reporting

1. Basic reporting
The writing and references are mostly clear and professional. I appreciate the detailed comparisons with similar studies in the introduction and discussion, including explanation of possible differences. The broad structure is also clear and most of the paper is 'self-contained'. However some aspects of structure and some figures and/or legends need improvement:

Major issues:
FIGURES: Most importantly, I don't see any figure supporting the conclusion (see review section 2 and 3)
WRITING/STRUCTURE: The statistics are not easy to comprehend as they are written now. I'm familiar with GLMM but it's not really clear what exactly has been done (see comments in experimental design)

Minor issues:
FIGURES:
Figure legends are sometimes incomplete or labels are missing:
-fig 1 please indicate what the grey arrows mean
-fig 4 scale on right y-axis is missing
-fig 5 The way I interpret the graph, it looks like B is the sustained signal and A the amplitude locked one. Please verify and adjust the legend accordingly
-fig 7 some of the markers are missing/double (2x empty squares, 2x empty round)
-fig 7&8 the comparison is shown between zf-zf for selectivity to CON and zf-cf selectivity to HET. I get the rationale, but what I would really like to see is a 4way-comparison. Therefore it would be clearer to include also selectivity of zf-zf to HET and zf-cf to CON
-fig 10 great to include this figure, Please also indicate unit for RS (#spikes/sec above baseline)
STRUCTURE:
1) The structure is mostly good but some parts are a bit long in my opinion and there is some redundancy. For example: 229-230 describes criterion, line 243 again. Please remove the first one because second is more complete
2) Also occasionally there are some (seeming?) contradictions in the text:
- I don't understand the parts in the data file and in discussion refering to inhibition, in the methods the authors write (line 238): "since units that show an inhibition as a result
of the auditory stimulus would not be included in the analysis
-676-677 the authors state that descending pathways from RAcup do not enter MLD, in line 695: "Since descending inputs to MLd originating in the RAcup ...". I suggest to leave out the latter, since it's part of section 'conlusion' whereas there is no data to conclude anything in this respect.

3) Not always self-contained: Occasionally, there is referred to a doctoral thesis including more detail. Please include that info in this manuscript or leave it out all together. For example, in line 318 t0 320 in the results section. In this case I would leave this out since it's accounted for by the GLMM
4) A bit more info on why the authors wanted to perform this study and what they expected would be helpful. Was there reason to reconsider findings by Woolley et al 04? Did they want to replicate the results? There is no hypothesis or expectation in the introduction.

Experimental design

Replication studies are important and MLd has not often been studied at all, so the scope of the paper is relevant to neuroscientists and behavioural biologists. The general idea behind the design, cross-fostering, is an elegant approach to answer the research question and has been used before. Most of the methods have been described properly with sufficient detail. However, I have concerns about experimental design and statistics.

Major issues
Experimental design:
1) If I understand correctly, in this study there is a possible confound in experimental design: the housing during development is different for the two groups. ZF-ZF group is housed indoor in small cages (line 128, however) whereas the ZF-CF group was raised in outdoor in aviaries (line 137). In line 124 authors suggests ZF-ZF were raised in indoor aviaries, but I assume these are the parents, not the juveniles. Please confirm.

Housing in outdoor aviaries may be considered as enriched environment (especially auditory enrichment due to other environmental sounds including more species), which has been shown to affect neural responsiveness (for example DOI:10.1155/2018/5903720). In addition, in aviaries birds get more exercise, which can also change basic neuronal processing (van Praag et al., 2000, DOI: 10.1038/35044558).
Since the strongest differences in this study is between responses overall, this is an alternative explanation for the results that should be seriously considered. The conclusion about the interaction effect between song type (CON/HET) and rearing condition on response strength is less convincing (see comments about figures and statistics), but even if this interaction is present, it could still be explained by an overall decrease in selectivity, resulting in a smaller difference in response between CON and HET in ZF-CF birds.

This doesn't necessarily mean auditory experience with Bengalese finch song does not affect neural development as well, the two are not mutually exclusive, but the current study cannot distinguish between the two explanations.
However, there are two possible solutions:
1) The authors could rephrase the discussion and conclusion and the data would still be relevant, new and worth publishing. The discussion and conclusion should include the above-mentioned alternative explanation. Future research can help disentangle
2) The authors perform a control experiment to potentially rule out the potential confound: raising additional zf-zf finches in the outdoor aviary and compare them with (preferably new) cage-raise zf-zf, and/or raise zf-cf in cages. However, I don't know if this is still feasible under the same conditions as this study.

2) Potentially major issue:
Were zf-cf birds housed in the same aviary as zf-zf birds? The conclusion seems to suggest so " ...cross fostered zebra finches were present even though ZF-ZF and ZF-CF birds were housed in the same aviary as adults..." (line 685-686) However, this is not clear from the methods. Otherwise there is another confound: CON is familiar to zf-zf but not zf-cf. Any difference could then also be explained by familiarity rather than experience during development. If in the same aviary please change line 144 to: ... birds were finally transferred to an indoor aviary at the Faculty of Medical and Health Sciences, where they were housed with the zf-zf birds. And change 172 to : ... therefore familiar to both groups of tested birds.

Minor issues
Insufficient for replication:
-Please include:
-statistical software and version (including package if applicable)
-material used for sound recording of the stimuli
-the cage size the zebra finches were reared in,
-usually zebra finches are also provided with other supplements than just water and food, please including this too

Validity of the findings

3. Validity of the Findings
As mentioned above, replication studies are important and data on MLd in songbirds is limited. However, the reason for replication is not clearly mentioned.
As far as I can determine (not my expertise) the electrophysiological measurements are performed correctly, sufficient animals have been tested and the general approach of statistics is good, avoiding pseudoreplication by nesting units within birds. However, I have my doubts about validity of the findings due to 2 possible confounds in the design, unclear statistics (SEE EXPERIMENTAL DESIGN) and importantly I don't see any graphs clearly supporting the conclusion.

Major issues:
1) Fig 10 is good to include but it doesn't not show a clear difference between reversed and forward song, nor CON vs HET. Therefore another figure would be helpful. For example one showing lines between CON and HET for each individual or each unit (the latter might be too much to be able to read the graph, the authors would have to try this out) with separate panels for zf-zf and zf-cf (see for example https://nl.mathworks.com/help/stats/repeatedmeasuresmodel.plot.html)

2) Statistics:
The use of GLMM is appropriate for the current type of study and appreciated since this is rare in electrophysiology studies! Nesting of the units within random factor 'subject' is a good choice to control for pseudoreplication. However I have my doubts about the implementation. I'm familiar with such models in R but it looks like other software has been used (please mention!), which may be part of the confusion.
The reasons I have my doubts is that some aspects are not described clearly enough to judge if it's correct, and the fact that no graphs clearly support the conclusion. Response strength in fig 10 for CON in zf-zf looks slightly higher but, I would not suspect a significant result. Also the low p-value for forward vs reversed song and little difference in fig 10 is worrying. This may be just a matter of writing but just to be sure, it would be good to verify where the p-values are coming from:
-If I remember correctly p-values for factors or interactions in the model itself are not meaningful if the model as a whole is not, because it only indicates the contribution too the model. For example, if significant effect for direction is found, but the model as a whole is not different from a null-model, the factor direction 'significantly contributes' to a null-effect. Is the final model compared to a null-model? Otherwise that might explain high F-values and low p-values that are not visible. For more info please find a comprehensible tutorial here: http://www.bodowinter.com/tutorials.html and/or verify with a statistician.

-It is incorrect to leave out sex as a variable, since the difference between CON and HET is different between sexes (thus interaction), the two can influence the results differently (line 304 - 306). The authors could include it as a random factor though.
-Were the assumptions for the model met? (homoscedasticity etc. see tutorial above)
-Please also explain more about the nesting of individual songs, I would include them as random factor nested within song types, but I'm not sure that's what was done here. Was this tested in a separate model?
-Have post-hoc tests been performed? If so and/or two models were used, were corrections for multiple comparisons been applied?

Minor issues:
Data file:
A (not raw) data file is included and clearly structured. Only a bit more info would be helpful: I would split up the first column into bird name and unit number. Please also include in the description that response strength BestFrequency are included. And what is the unit of BF here? Does it indicate the actual frequency tone in kHz or the response strength?

I can imagine that raw recordings are too large to share, but the authors indicate that other data will be shared on other platforms. At what level of processing will this data be shared?

Additional comments

Apologies for the lengthy review! I hope you can take away some of my worries. Worst-case scenario, there is no difference between ZF-ZF and ZF-CF in response to CON and HET. In that case the data are still interesting and worth publishing because there is a clear general difference between ZF-ZF and ZF-CF MLd responsiveness and this replicates earlier findings by Woolley et al.2010. However, the data would be inconclusive as to what causes the difference: housing, tutoring or familiarity.

---

## Round 0.2 · Minor Revisions

Please address the minor revisions suggested by reviewer 2. I want to particularly point out the discrepancy between your pdf file and the tracked changes file - as pointed out by reviewer 2 please make sure your final file contains all changes made. I am looking forward to receiving your revised manuscript.

Reviewer 1 ·

Basic reporting

The revisions have addressed all of my concerns about basic reporting and I especially appreciate the comprehensive explanation of their use of GLMMs. This type of model should be used more extensively in the field and I think the methodological description provided will facilitate the use of GLMMs by other researchers.

Experimental design

I think the experimental design is sound. Reviewer 2 raised a valid concern with regard to a confound between the impact of overall rearing environment (indoor/outdoor rearing) and the specific experience of home rearing vs. cross fostering. I think that the wording in the current draft sufficiently acknowledges this confound and also serves to prevent over-interpretation of the results.

Validity of the findings

The changes made to the analysis, especially the significant effort made to improve the GLMM analysis, have satisfied my concerns about validity of the primary findings.

Reviewer 2 ·

Basic reporting

Apologies for the late reply.
The manuscript has greatly improved. Most of my questions have been answered satisfactorily. The new graphs that were added clearly contribute to the understanding of the manuscript and are now convincing. Thanks for also sharing the raw data.

Basic reporting
-The d-prime equation is missing a variance sign before RSb, apostrophe in wrong place. And shouldn’t RSa and RSb be meanRSa and mean RSb? (as in Doupe and Solis97?)
-Regarding the explanation below the equation: technically, d-prime is the discriminability/selectivity (not difference)

-There are a few difference between the track changes file the pdf. Two I notices:
-in doc (not pdf) there is still 2 paragraphs on response strength
-line 524 not complete in doc
There might be more so be careful with which version you work for the final version

-line 340 equation u in Zu not explained. Should be “ related to a fixed u” I assume?
Even more helpful would be something like:
X includes: Bird category, song type (CON/HET), direction (FOR/REV), SongID, Bird Category x CON/HET
Z includes factors: bird ID and unit within bird
Or alternatively something like this (I know I’m biased to R notation, alternatives are fine)
model<- glmm (Bird category * song type + direction + songID, random = 1|birdID/unit)
where / means within bird

-line 370 HET=3R typo --> HET-3

Experimental design

Experimental design

The authors also explained the methods better, now one potential confound (familiarity) can be excluded. I appreciate they are more carefull with wording regarding rearing conditions vs cross-fostering, however I would prefer an explicit mention of this (shortly, one or two sencences is enough). You could use your own wording in the response letter for example “ our data show a difference in response to the auditory stimuli as a result of rearing conditions – but we make no attempt at identifying which of the variables creates the difference between the two group of birds” or “ our data show a difference in response to the auditory stimuli as a result of rearing conditions, but we are not attributing the differences specifically to cross-fostering”
I think it’s more scientific to be open about this and it doesn’t devalue your study in any way. It is highly interesting or relevant.

Validity of the findings

Validity
I also appreciate the effort of explaining and improving the statistics and this looks convincing, only a few minor clarifications would be helpful.

-The paragraph on assumptions was a bit confusing to me: were normality and levene’s test tested on risiduals? To my understandig is generalized linear model per definition not normally distributed and assumptions are about risduals, if I remember correctly (the log transformation is performed by the link function in R, not sure how it woks in SAS)? (I would also find it hard to believe that Levene’s test is not significant, given the difference in variance visible in figure 10). Was heteroscedasticity excluded?

-Model comparison seems good, maybe explicitly mention null model comparison

-Table 5, why are there 2 t-values and 2 p-values for the LS means, aren’t comparisons between groups? (i.g. ZF-ZF CON vs ZF-ZF HET)?

Additional comments

I appreciate all the work that has been done and the manuscript really improved.

---

## Round 0.3 · accepted · Accept

You have addressed all concerns of the reviewers. Please inspect the proofs carefully once you receive them, checking for the formerly raised concerns with conversion to pdf.